# Revealing *mitf* functions and visualizing allografted tumor metastasis in colorless and immunodeficient *Xenopus tropicalis*
Rensen Ran [1,2,4] ✉, Lanxin Li[1,4], Tingting Xu[3], Jixuan Huang[1], Huanhuan He [2] & Yonglong Chen [1] ✉

Transparent immunodeficient animal models not only enhance in vivo imaging investigations of visceral organ development but also facilitate in vivo tracking of transplanted tumor cells. However, at present, transparent and immunodeficient animal models are confined to zebrafish, presenting substantial challenges for real-time, in vivo imaging studies addressing specific biological inquiries. Here, we employed a $mitf^{-/-}/prkdc^{-/-}/il2rg^{-/-}$ triple-knockout strategy to establish a colorless and immunodeficient amphibian model of *Xenopus tropicalis*. By disrupting the *mitf* gene, we observed the loss of melanophores, xanthophores, and granular glands in *Xenopus tropicalis*. Through the endogenous *mitf* promoter to drive BRAF$^{V600E}$ expression, we confirmed *mitf* expression in melanophores, xanthophores and granular glands. Moreover, the reconstruction of the disrupted site effectively reinstated melanophores, xanthophores, and granular glands, further highlighting the crucial role of *mitf* as a regulator in their development. By crossing $mitf^{-/-}$ frogs with $prkdc^{-/-}/il2rg^{-/-}$ frogs, we generated a $mitf^{-/-}/prkdc^{-/-}/il2rg^{-/-}$ *Xenopus tropicalis* line, providing a colorless and immunodeficient amphibian model. Utilizing this model, we successfully observed intravital metastases of allotransplanted xanthophoromas and migrations of allotransplanted melanomas. Overall, colorless and immunodeficient *Xenopus tropicalis* holds great promise as a valuable platform for tumorous and developmental biology research.

Transparent animal models are invaluable for conducting precise in vivo imaging investigations in the realms of developmental biology and tumor studies[1]. Immunodeficient animal models, in particular, prove beneficial for addressing various biological inquiries, including the in vivo tracking of transplanted tumor cells[1,2]. Transparent immunodeficient animal models, beyond their facilitation of in vivo imaging studies pertaining to organ development, also provide essential support for the dynamic tracking and investigation of transplanted tumor cells[3,4]. However, to the best of our knowledge, the current scope of transparent immunodeficient animal models is confined to the zebrafish model. The use of zebrafish introduces substantial challenges for addressing biological inquiries such as real-time, in vivo imaging studies of lung development, orthotopic transplantation of lung cancer, and the progression of lung cancer, in addition to issues like genetic redundancy[5–8]. *Xenopus tropicalis* is a well-established, diploid amphibian animal model that has been extensively utilized in

developmental genetics and tumor biology fields[9,10]. Despite its utility, to the best of our knowledge, reports of transparent and immunodeficient *Xenopus tropicalis* have yet to be published[11–14]. Hence, the transparent glassfrog serves as an inspiration for us to develop another transparent and immunodeficient amphibian model in *Xenopus tropicalis*, thereby diversifying the range of transparent immunodeficient animal models. This will enable future utilization of in vivo imaging in transparent immunodeficient frogs to address questions that may not be easily answered in the zebrafish model[15].

Presently, pigment cells identified in mammalian skin exclusively comprise melanocytes. With the exception of the retinal pigment epithelium, pigment cells in vertebrates derive from neural crest cells, demonstrating a substantial degree of developmental conservation[16,17]. In contrast, the skin of fish and amphibians typically encompasses three primary types of pigment cells: melanophores (named melanocytes in mammals), xanthophores, and iridophores[18]. These cutaneous pigment cells exert a

[1]Department of Chemical Biology, Guangdong Provincial Key Laboratory of Cell Microenvironment and Disease Research, Shenzhen Key Laboratory of Cell Microenvironment, School of Life Sciences, Southern University of Science and Technology, 518055 Shenzhen, China. [2]Guangdong Provincial Engineering Research Center of Molecular Imaging, The Fifth Affiliated Hospital of Sun Yat-sen University, 519000 Zhuhai, China. [3]Fujian Medical University Union Hospital, 350001 Fuzhou, China. [4]These authors contributed equally: Rensen Ran, Lanxin Li. ✉e-mail: sankeshumy@163.com; chenyl@sustech.edu.cn

 1

profound influence on skin transparency[12,13,19]. In *casper* zebrafish, transparent skin arises from the absence of both light-absorbing melanophores and reflective iridophores, which is associated with the double mutant of *mitfa* and *mpv17*[4,20,21]. *MPV17* is a human gene encoding a mitochondrial inner membrane protein (MPV17), and its malfunction or loss can cause rare autosomal recessive disorders termed mitochondrial DNA depletion syndromes[22]. Adult mice with homozygous mutations in the MPV17 gene developed nephrotic syndrome and chronic renal failure, with a survival rate of less than 10% at six months[23]. In zebrafish, the orthologous gene of human *MPV17* is *transparent* (*tra*), also known as *mpv17*, and its mutations can lead to complete loss or significant reduction of iridophores in zebrafish[21,24]. Nonetheless, the precise contribution of *mpv17* towards the transparent skin phenotype in *casper* zebrafish still requires further investigation[25]. In *Xenopus tropicalis*, mutants were engineered through *tyr* knockout, *slc2a7* knockout, or a triple knockout involving *tyr*, *slc2a7*, and *hps6*, aiming to develop a transparent frog. Nonetheless, challenges arise from the impact of granule glands on skin transparency and heightened mortality rates in the offspring[14]. Considering these factors, and to ensure optimal survival rates and minimal impact on health in the final construction of transparent immunodeficient frogs, we have chosen to initiate model development by attempting to knock out the *mitf* gene in *Xenopus tropicalis*.

Mutated *mitfa* has been unequivocally found to cause a depletion of light-absorbing melanophores in *casper* zebrafish, which is a crucial aspect leading to the skin transparency[4,20]. *Mitf* is the mammalian ortholog of zebrafish *mitfa* and has homologs across species, ranging from primitive metazoans to higher-order mammals. *Mitf* encodes a transcription factor, microphthalmia-associated transcription factor (Mitf), which belongs to the MiT family and contains a basic helix-loop-helix leucine zipper (bHLH-Zip) DNA binding and dimerization domain[26]. In mammals, a wide variety of Mitf isoforms have been identified, and express widely or in a tissue-specific manner[26]. In *casper* zebrafish, the mutant *mitfa*[113stop] is instrumental in rendering the skin transparent by disrupting Mitf's regulatory role in melanophore development, thus preventing skin from melanin pigmentation[20]. Of note, *mitfa* is also expressed in zebrafish xanthophores, yet its function in xanthophore development remains uncertain[27,28]. While there are no reports investigating *mitf* in *Xenopus tropicalis*, limited data from *Xenopus laevis* suggests that this gene could act as a master regulator of melanophore development in *Xenopus tropicalis*[29–32]. Therefore, disrupting *mitf* presents an attractive opportunity to establish transparent *Xenopus tropicalis*.

Up till now, intravital imaging of engrafted tumor cells and therapy response studies have established that transparent and immunodeficient zebrafish is a first-class animal model[3,5]. In zebrafish, an ample level of immunodeficiency for tumor transplantation can be attained by disrupting *prkdc* to impair T and B lymphocyte development, and mutating *il2rg* to eliminate mature natural killer (NK) cells[3]. *PRKDC* is an essential gene for the DNA-dependent protein kinase (DNA-PK) complex, which consists of Ku80 (encoded by *XRCC5*), Ku70 (encoded by *XRCC6*), and the catalytic subunit of DNK-PKcs (encoded by *PRKDC*)[33]. This complex plays a crucial role in repairing double-strand breaks in genomic DNA and facilitating V(D)J recombination in the T and B lymphocytes of mammals and amphibians alike[34,35]. *IL2RG* encodes IL2RG (CD132), which is a shared receptor subunit for six cytokines (IL-4, IL-7, IL-9, IL-15 and IL-21), plays a vital role in lymphoid development by activating JAK3 and is expressed on the surface of T, NK, NKT and dendritic cells[36]. In *Xenopus tropicalis*, tumor transplantations are feasible in thymectomized or sub-lethally gamma-irradiated or *rag2*[−/−] individuals that serve as recipients[11,37]. Other immunodeficient *Xenopus tropicalis* models have been developed via the knockout of *XNC10.1* or *foxn1*, but their suitability for tumor transplantation is yet to be validated[38,39]. Therefore, we envisioned the generation of a transparent and immunodeficient frog through the dual disruption of *prkdc* and *il2rg*, building upon the foundation of *mitf* null *Xenopus tropicalis*.

Eventually, through the founder generations of *mitf*[−/−] and *prkdc*[−/−]/*il2rg*[−/−] frogs, the *mitf*[−/−]/*prkdc*[−/−]/*il2rg*[−/−] triple-knockout *Xenopus tropicalis* line was established. Actually, this frog line

resulted in a colorless and immunodeficient amphibian animal model of *Xenopus tropicalis*, making it a suitable recipient for transplantation of xenogeneic skin, allogeneic xanthophoroma, and melanoma. Moreover, knocking out *mitf* in *Xenopus tropicalis* revealed previously unknown functions of *mitf* in the development of granular glands and xanthophores. Collectively, our *mitf*[−/−]/*prkdc*[−/−]/*il2rg*[−/−] *Xenopus tropicalis* line presents a valuable amphibian tool for intravital investigations of tumorous and developmental biology.

## Results

### Biallelic *mitf* disruption creates colorless *Xenopus tropicalis*

Due to the transparency of skin in *Xenopus tropicalis* is influenced by light-absorbing melanophores, necessitating the targeting of *mitf* to impede melanophore development. In recognition of the intricate transcriptional regulation of the *mitf* locus[26], a set of guide RNAs were tailored to specifically target exons of *mitf* to knockout *mitf* (Supplementary Fig. 1 and Supplementary Fig. 2a). All guide RNAs effectively induced damage to their respective targets, however, melanophore loss was only observed in the mosaic founder generations that were targeted by gRNA T5 or gRNA T7 (Supplementary Fig. 2b, c, Supplementary Fig. 3a). As shown in Supplementary Fig. 1, the target site for gRNA T5 was located upstream of the core domain of Mitf, while the target site for gRNA T7 was located in the helix2 domain of Mitf. Therefore, G0 founder tadpoles (79/108) generated by gRNA T7, which exhibited a noticeable lack of melanophores, were selected for further experimentation (Fig. 1a). To verify the genotype-phenotype specificity of SpCas9 *mitf* site of gRNA T7, genomic material from ten randomly assigned melanophore-phenotypic tadpoles was extracted and performed off-target analyses. Using CRISPOR[40], we determined the genome-wide potential off-target sites and selected those with no more than five base mismatches (Supplementary Data 1). Upon further inspection, we found that none of the off-target sites containing four or fewer mismatches exhibited any detectable mutations (Supplementary Fig. 3b and Supplementary Data 2). Consequently, these findings suggested off-target effects were infrequent in the tadpoles examined.

Mosaic tadpoles generated via gRNA T7 exhibited a surprising transparency of the skin and made certain internal organs externally visible beyond one-month postmetamorphosis (Fig. 1b). The mosaic frogs (G0 founders) were subsequently raised to sexual maturity and their offspring (F1 generations) obtained through reciprocal breeding of G0 founders. Of the F1 tadpoles, 46% displayed normal melanophore development without any observable differences, while the remainder showed melanophore disappearance (Supplementary Fig. 4a). From this F1 generations, ten tadpoles exhibiting normal melanophores or fifteen offspring with melanophore loss were randomly selected and subjected to genotyping. Seven offspring with melanophore-present exhibited heterozygous genotypes, while the remaining melanophore-present offspring were of the wild type (Supplementary Fig. 4b). All 15 melanophore-loss F1 offspring exhibited compound mutations of biallelic *mitf*, indicating a complete loss of Mitf function in these melanophore-loss offspring (Supplementary Fig. 4c). Individuals from the F1 generation carrying both alleles of the *mitf* disruptive mutation, which can cause frameshift mutations, were selected for further breeding to obtain the F2 generation (Supplementary Fig. 4d). Hereinafter, mutants displaying a functional null of biallelic *mitf* were referred to as *mitf*[−/−] individuals, and those exhibiting a functional null of monoallelic *mitf* as *mitf*[+/−] individuals.

The F1 *mitf*[−/−] *Xenopus tropicalis* tadpoles solely possessed melanin pigmentation in the eyes, with a lack of pigmentation observed in the other part of body, resulting in the transparent skin, in addition to some residual reflective cells in the peritoneum visible post-stage 48 (Fig. 1c and Supplementary Fig. 5). After metamorphosis, the F1 *mitf*[−/−] frog skin became transparent within a two-month period, enabling clear visualization of internal organs such as the liver and lungs from an external perspective (Fig. 1d, e). Beyond this period, however, the F1 *mitf*[−/−] frog skin turned colorless and opaque, impeding external visualization of the internal organs (Fig. 2a). To compare skin transparency among adult wild type, *mitf*[−/−], and

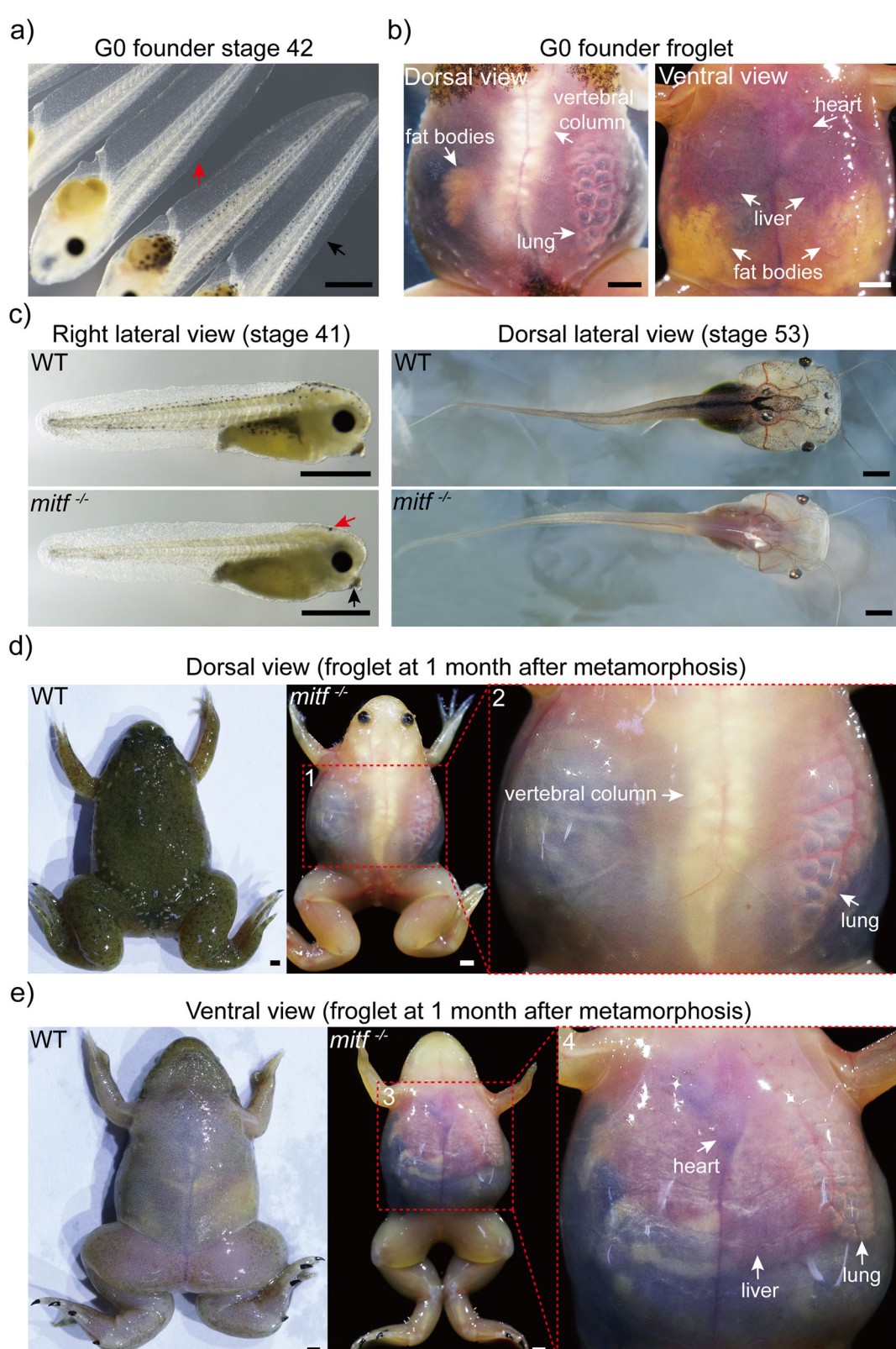

The results revealed that the exclusion of blood had a negligible effect on the skin transparency of *Xenopus tropicalis* (Fig. 2c). Moreover, we conducted a comprehensive examination of melanophores throughout the entire body of adult *mitf*[−/−] *Xenopus tropicalis*. We found a lack of black pigmentation in most internal organs, with the exception of melanomacrophage centers in the lungs[42], black claws resulting from modified corneocytes of the corneous

*tyr*[−/−] *Xenopus tropicalis*[41], their skin transparency was evaluated for light transmission. Figure 2b clearly indicated that the skin of *mitf*[−/−] *Xenopus tropicalis* exhibited the highest degree of light transmission, enabling a clear view of the Chinese characters "理想" behind it. Given the potential impact of blood on the transparency of amphibian skin[15], we examined the influence of blood on the transparency of adult *mitf*[−/−] *Xenopus tropicalis* skin.

**Fig. 1 | Biallelic *mitf* disruption makes *Xenopus tropicalis* transparent within two months post-metamorphosis. a** After using the CRISPR/Cas9 system to knock out *mitf* via gRNA T7, founder generation (G0) mosaic tadpoles at stage 42 exhibited a loss of melanophores (indicated by red arrow), compared to wild type (WT) tadpoles from the same batch (indicated by black arrow). 79 mosaic tadpoles and 20 wild type tadpoles were observed. **b** One month after metamorphosis, the transparent skin of the mosaic froglet allowed for external visibility of internal organs such as the fat bodies, lung, liver, heart, and vertebral column. Here was shown one representative froglet, out of a total of 11 mosaic froglets. **c** The F1 *mitf*[−/−] *Xenopus tropicalis* tadpole exhibited melanophores loss throughout the entire body at stages 41 and 53 compared to wild type tadpoles from the same batch. Notably, at stage 41, the melanin pigmentation in the eyes of *mitf*[−/−] *Xenopus tropicalis* was generated by retinal pigment epithelium (RPE) cells, and redistribution of melanin in oocytes resulted in

some melanin pigment in the head (indicated by red arrow) and cement gland (indicated by black arrow). However, the melanin pigment in the head and cement gland disappeared in later development stages, such as stage 53. Representative photographs were exhibited from 35 F1 *mitf*[−/−] *Xenopus tropicalis* tadpoles at stage 41, 15 F1 *mitf*[−/−] *Xenopus tropicalis* tadpoles at stage 53, as well as from 10 wild type tadpoles each at stage 41 and 53. **d, e** One month after metamorphosis, the F1 *mitf*[−/−] *Xenopus tropicalis* froglet exhibited transparent skin from both dorsal (**d**) and ventral (**e**) views. Red dashed boxes 2 and 4 corresponded to magnified views of red dashed boxes 1 and 3, respectively. The transparent skin allowed for external visibility of internal organs including the lung, liver, heart, and vertebral column within one month after metamorphosis as shown. Ten F1 *mitf*[−/−] *Xenopus tropicalis* froglets and ten wild-type *Xenopus tropicalis* froglets were used to provide representative photographs as shown. Each scale bar is 1 mm.

layer[43], and black eyes due to retinal pigment epithelium cells[32] (Supplementary Fig. 6). Importantly, F1 *mitf*[−/−] *Xenopus tropicalis* exhibited normal growth and reproductive behavior, and we easily generated the F2 generation through germline transmission of F1 frogs. Furthermore, a noticeable difference in skin color was observed between *mitf*[−/−] and *tyr*[−/−] *Xenopus tropicalis*, which lacked melanin pigmentation due to the inability to biosynthesize melanin[41,44], indicating possible alterations in other pigment cell types in *mitf*[−/−] *Xenopus tropicalis* (Fig. 2a).

Together, by functionally nullifying biallelic *mitf* mutations, we've successfully created a colorless line of *Xenopus tropicalis*. This novel line displays transparent skin and visible internal organs within two months postmetamorphosis, and lacks cutaneous melanophores throughout its lifespan. This newly developed amphibian model holds great potential for use in tumorous and developmental studies[3,4].

## Crucial role of *mitf* in melanophore, xanthophore, and granular gland developments in *Xenopus tropicalis*

*Mitf* plays a crucial role as a master regulator in the development of melanocytes across mammals and fish[26]. Loss or malfunction of *Mitf* results in the absence of melanocytes in these species. To confirm the absence of melanophores in *mitf*[−/−] frogs, skin and eye samples of *mitf*[−/−] and wild type *Xenopus tropicalis* were subjected to histopathological observation. Unexpectedly, the skin of *mitf*[−/−] *Xenopus tropicalis* exhibited a concurrent absence of melanophores and granular glands (Fig. 3a). The loss of melanophores was also confirmed in the choroid of *mitf*[−/−] *Xenopus tropicalis* (Fig. 3b). Moreover, the absence of both granular glands and melanophores was confirmed through transmission electron microscopy (TEM) analysis of skin and eye samples of *mitf*[−/−] and wild-type *Xenopus tropicalis* (Fig. 3c–h and Supplementary Fig. 7). Furthermore, through the characterization of the granular gland development process in both wild type and *mitf*[−/−] *Xenopus tropicalis*, we observed that gland rudiments in the skin of *mitf*[−/−] *Xenopus tropicalis* may undergo normal development at stage 58, but mature granular glands fail to emerge at stage 60 (Supplementary Fig. 8). Notably, throughout the complete developmental stages of *mitf*[−/−] *Xenopus tropicalis*, xanthophores were notably absent, alongside the absence of granular glands in the entire organism (Fig. 3c-h and Supplementary Fig. 9).

As regards the granular glands of *Xenopus tropicalis*, they share numerous structural and functional similarities with mammalian exocrine sweat glands[45], including similar tissue architecture enveloped by myoepithelial cells[46], expression of the classical aquaporin Aqp5 that regulates water homeostasis and other physiological processes[47], and secretion of antimicrobial peptides, such as Nmb and Pgq, for safeguarding the skin against pathogenic microorganisms[48,49] (Supplementary Fig. 10). Nonetheless, to the best of our knowledge, the involvement of *Mitf* in the development of granular or exocrine sweat glands has yet to be reported[45,46]. Hence, it is imperative to conduct a meticulous examination of the impact of mutated MITF on the formation of sweat glands in mice and humans, in light of the findings from *mitf*[−/−] *Xenopus tropicalis*. In terms of xanthophores in *Xenopus tropicalis*, their gene regulation network and developmental pathways have not been extensively investigated in comparison to

their zebrafish counterparts[13,50]. Additionally, there are no reports on the role of *mitf* in xanthophore development in any species to date[26,51]. Nevertheless, given the intricate transcriptional regulation and functional diversity of the *mitf* locus[26], it is plausible that *mitf* plays a crucial role in the development of xanthophores and granular glands in *Xenopus tropicalis*, as indicated by the phenotype of *mitf*[−/−] *Xenopus tropicalis*. Therefore, subsequent studies were necessary to firmly establish the genotype-phenotype specificity of *mitf* in relation to xanthophores and granular glands in *Xenopus tropicalis*.

To uncover the disruption of melanophore development in *mitf*[−/−] *Xenopus tropicalis*, we analyzed the expression of *mitf* and melanophore markers (*tyr*, *pmel*, *dct*, and *tyrp1*) using whole mount in situ hybridization[26]. Our results showed that the observed changes in marker gene expression in early embryonic melanocytes of *Mitf-mutated* mice and zebrafish were also apparent in *mitf*[−/−] *Xenopus tropicalis*[20]. Specifically, melanoblasts labeled by *dct*, *pmel*, and *tyrp1* were detected, but in a substantially reduced number (Fig. 4a and Supplementary Fig. 11–20). Notably, we observed a persistent absence of mature melanophore marker gene (*tyr*)-labeled cells in *mitf*[−/−] *Xenopus tropicalis* (Fig. 4a and Supplementary Fig. 14), underscoring the essential role of *mitf* in the maintenance and survival of melanoblasts in *Xenopus tropicalis*, as in mammals and fish[26,52]. Concomitantly, reverse transcription-polymerase chain reaction (RT-PCR) analysis further confirmed the evident reduction or absence of melanophore markers (*tyr*, *pmel*, *mlana*, *ednrb1*, *slc45a2*, and *dct*) in the skin of *mitf*[−/−] individuals compared to that of wild-type *Xenopus tropicalis* (Fig. 4b). Therefore, the whole-mount in situ hybridization and RT-PCR results suggest a total absence of mature melanophores throughout the body of *mitf*[−/−] *Xenopus tropicalis*. This is likely attributed to the insufficient Mitf activity in melanoblasts, posing challenges to their survival.

Given the absence of xanthophores and granular glands in *mitf*[−/−] *Xenopus tropicalis*, we investigated whether there were changes in gene expression related to these features. As there were limited literatures on xanthophore-related genes in this species, we used *Xenopus tropicalis* homologs of zebrafish xanthophore-related genes (*trpm1*, *pax3*, and *gjd2*) for RT-PCR analysis[53]. The expression of these genes in the skin of *mitf*[−/−] *Xenopus tropicalis* was evidently lower compared to the wild type, suggesting an abnormality in xanthophore development in these frogs (Fig. 4b). To corroborate the loss of granular gland phenotype in *mitf*[−/−] *Xenopus tropicalis*, genes pertinent to granular gland function, including aquaporins (*aqp5* and *aqp8*)[47], Ca²⁺-activated chloride channel anoctamin 1 (*ano1*)[45,54], antimicrobial peptides (*nmb* and *pgq*)[48,49], and hormones (*trh*, *xt6l*, and *tph1*)[55], were selected and subjected to RT-PCR analysis. Notably, *Aqp5* and *Ano1* are established markers for exocrine sweat glands in some mammals. As expected, the skin of *mitf*[−/−] *Xenopus tropicalis* exhibited evidently reduced expression of all these genes (Fig. 4c–e). Together, the molecular evidence provided here also suggests abnormal developments of xanthophores and granular glands (Fig. 4).

To strengthen the genotype-phenotype specificity of *mitf*[−/−] *Xenopus tropicalis*, we conducted a rescue experiment by integrating a foreign DNA fragment into the disrupted *mitf* locus to repair the damaged site (Fig. 5a).

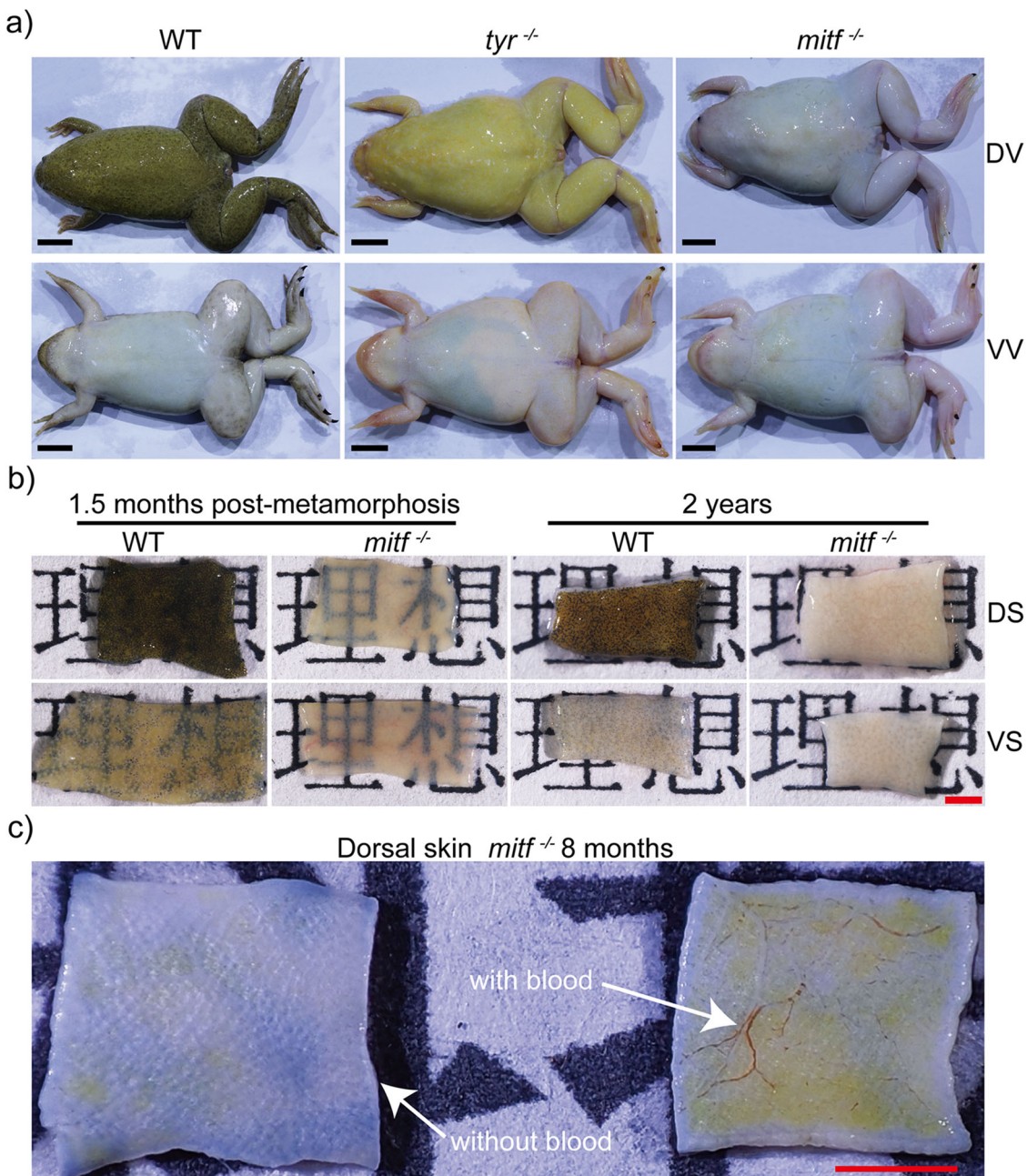

**Fig. 2 | The skin of** *mitf*$^{-/-}$ **adult frogs turns colorless and opaque. a** The representative photographs of adult wild type (WT), *tyr*$^{-/-}$, and *mitf*$^{-/-}$ *Xenopus tropicalis* were displayed to showcase their dorsal view (DV) and ventral view (VV). The photographs were obtained from a sample of 15 adult frogs of each genotype, all of which were 1 year old. **b** Skin samples approximately 2 mm×5 mm in size were obtained from both dorsal skin (DS) and ventral skin (VS) of wild-type and *mitf*$^{-/-}$ *Xenopus tropicalis* aged 2 years or 1.5 months postmetamorphosis. These samples were utilized in transillumination experiments to assess their translucency properties. Specifically, the experiment involved positioning each sample on a sheet of paper with the word "理想" and photographing them under consistent lighting and camera settings to assess the visibility of the word. Three frogs were used for each genotype, and one sample of DS and VS skin was collected per frog. Representative photographs of each genotype were presented to illustrate the experimental findings. **c** Skin samples were obtained with blood or after exlusion of blood from dorsal regions of the same *mitf*$^{-/-}$ *Xenopus tropicalis* aged 8 months. Three frogs were used for this experiment. The representative photograph was presented. The black scale bar in **a** is 1 cm. The red scale bar in **b** and **c** is 1 mm.

Concretely, we inserted a foreign DNA fragment that included the final two exons and the 3′UTR region of the *mitf* locus. This successfully repaired the damaged *mitf* site in *mitf*$^{-/-}$ *Xenopus tropicalis*, as confirmed by the diverse degrees of melanophore rescue phenotypes observed in the mosaic *Xenopus tropicalis* (Fig. 5b-c). In addition, xanthophores were also restored in the melanocytes-bearing skin of the mosaic frog (Fig. 5c). Although we didn't find mature granular glands in the rescued skin of mosaic frogs, early-stage granular glands[56] were detected in the rescued skin (Fig. 5d). The lack of developed mature granular glands in the rescued skin of mosaic frogs is

likely closely related to their mosaic nature. Consequently, these findings further corroborated the genotype-phenotype specificity of the melanophores, xanthophores, and granular glands loss caused by *mitf* knockout in *Xenopus tropicalis*.

Coincidentally, as shown in Fig. 6a, we were also utilizing the strategy of the specific integration of human *BRAF*$^{T1799A}$ into the *mitf* locus to establish a *Xenopus tropicalis* melanoma model. This model happened to address the issue of the lack of specific and reliable Mitf antibodies for detecting *mitf* expression in *Xenopus tropicalis*. *BRAF*$^{T1799A}$ is a frequently occurring point

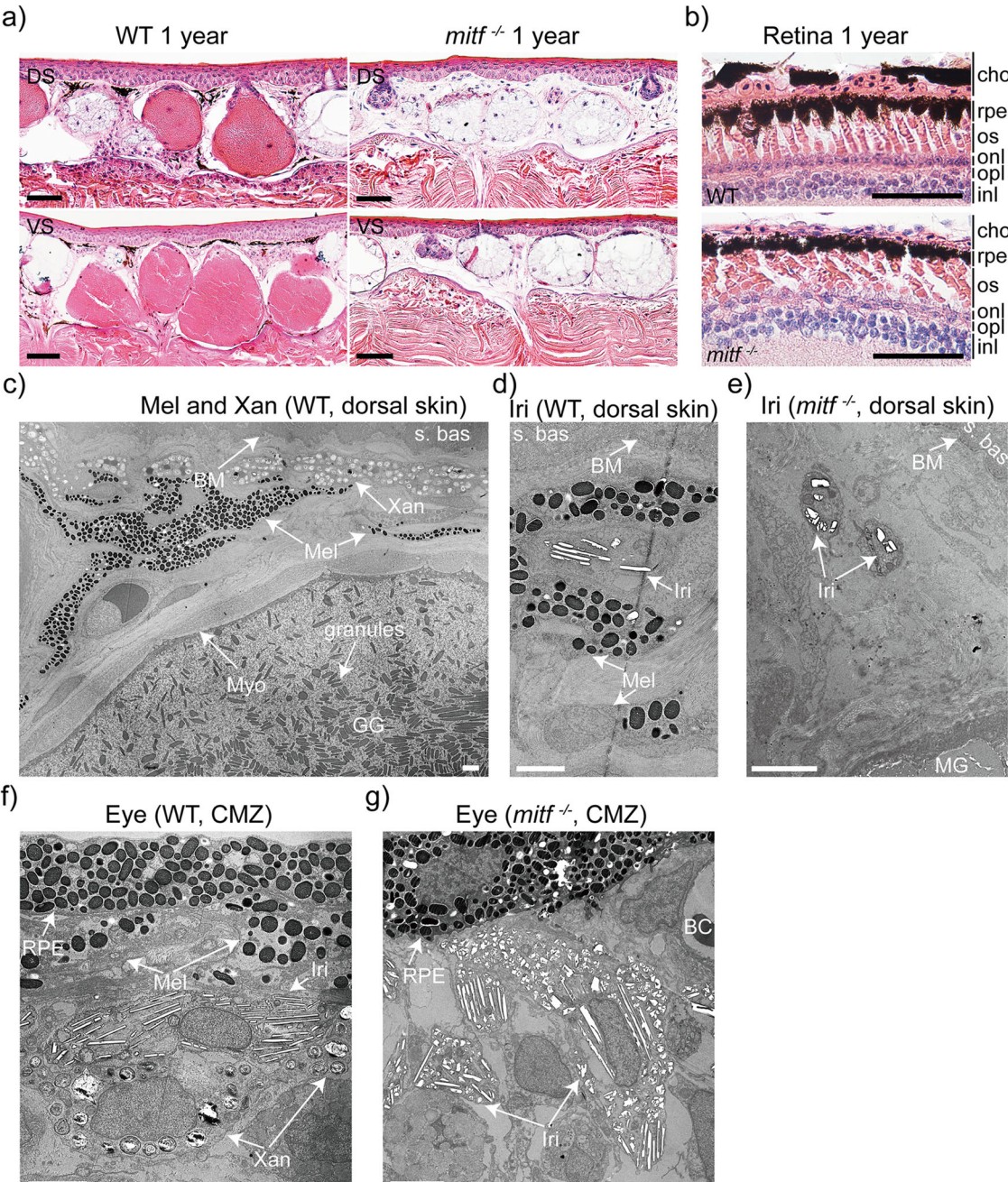

**Fig. 3 | Melanophores, xanthophores, and granular glands were absent in *mitf*<sup>−/−</sup> *Xenopus tropicalis*. a** Histological structure representative of the dorsal (up line) and ventral (bottom line) skin of 1-year-old WT and *mitf*<sup>−/−</sup> *Xenopus tropicalis* was displayed. **b** The histological structure representative of the retina in eyes of 1-year-old WT and *mitf*<sup>−/−</sup> *Xenopus tropicalis* was shown. cho, choroid; rpe, retinal pigmented epithelium; pl, photoreceptor layer of rods and cones; onl, outer nuclear layer; opl, outer plexiform layer; inl, inner nuclear layer; ipl, inner plexiform layer; gcl, ganglion cell layer. Five frogs of each genotype underwent H&E staining. Two dorsal skin samples, two ventral skin samples, and both eyes were collected from each frog. Histological examination of each collected sample was conducted on 10 paraffin sections. **c, d** The transmission electron microscopy (TEM) examination of the dorsal skin of 1-year-old WT *Xenopus tropicalis* was exhibited. The results revealed the presence of melanophores and xanthophores in the dermis (**c**), and the coexistence of iridophores and melanophores (**d**). **e** The TEM examination of the

dorsal skin of 1-year-old *mitf*<sup>−/−</sup> *Xenopus tropicalis* was exhibited. **f, g** The TEM examination of the ciliary marginal zone in retina of 1-year-old WT and *mitf*<sup>−/−</sup> *Xenopus tropicalis* was exhibited. The results revealed the presence of melanophores, xanthophores, and iridophores in the WT ciliary marginal zone (**f**), while melanophores and xanthophores were absent in the *mitf*<sup>−/−</sup> ciliary marginal zone (**g**). TEM analysis was performed on three frogs of each genotype. For each frog, three 1 mm×1 mm skin samples were collected from distinct regions on the dorsal skin, and one eye sample was randomly selected. Five ultrathin sections were generated from each sample for generating the representative results presented here. Mel, melanophore; Xan, xanthophore; Iri, iridophore; BM, basement membrane; MG, mucous gland; GG, granular gland; Myo, myoepithelial cell; s.bas, stratum basale; s.spi, stratum spisosum; s.gra, stratum granulosum; s.cor, stratum corneum; BC, blood cell; CMZ, ciliary marginal zone. The black scale bar in **a–b** is 50 μm. The scale bar in **c–g** is 2 μm.

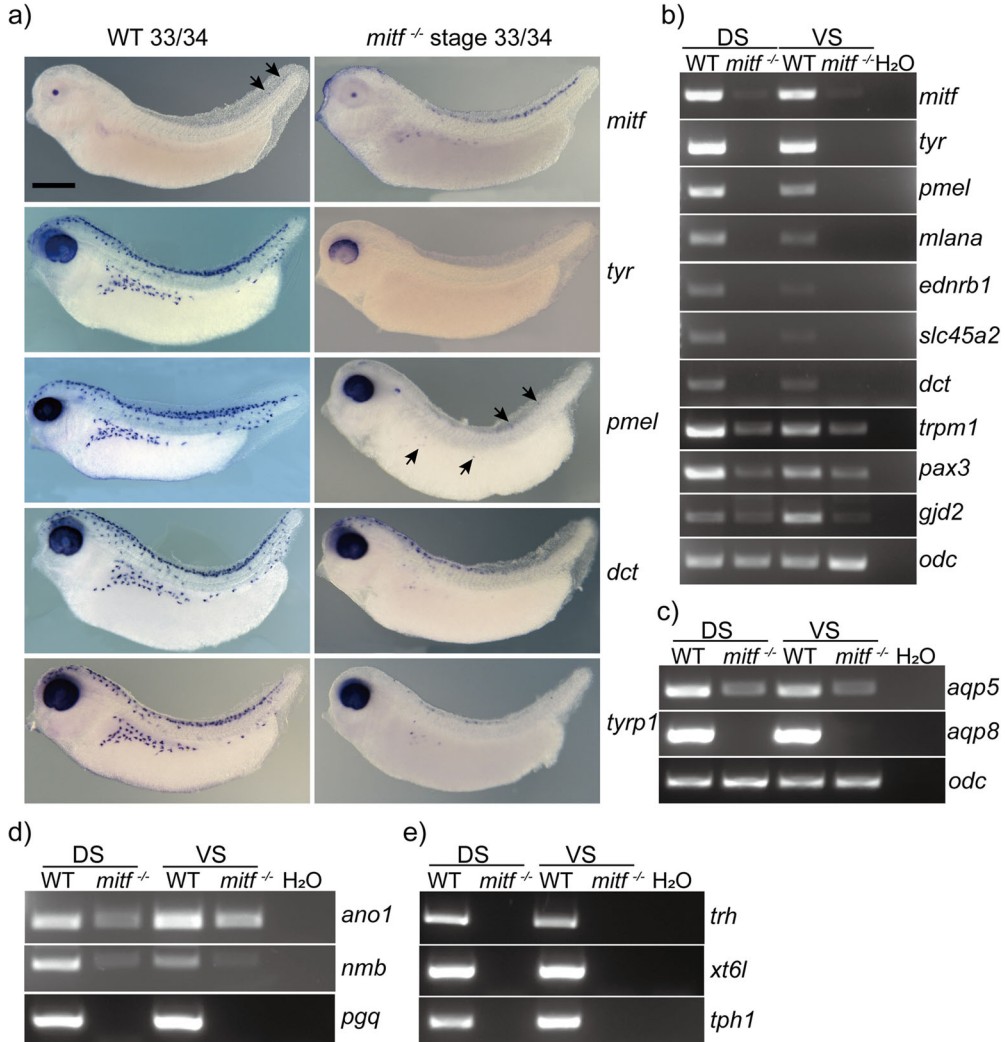

**Fig. 4 | The expression of crucial genes associated with melanophores, xanthophores, and granular glands was reduced or lost in *mitf*⁻/⁻ *Xenopus tropicalis.*** **a** Whole-mount in situ hybridization analysis of stage 33/34 *Xenopus tropicalis* embryos demonstrated a reduction or absence of mRNA hybridization signals in *mitf*⁻/⁻ embryos compared to the WT. The black arrows indicated some weaker mRNA hybridization signals. At least five embryos were used for in situ hybridization of each gene in every genotype embryo. **b–e** Representative results of RT-PCR analysis for several melanophores-related genes, including *mitf*, *tyr*, *pmel*, *mlana*, *ednrb1*, *slc45a2*, and *dct*, as well as xanthophores-related genes, *trpm1*, *pax3*, and *gid2*, were presented in **b**. The expression of aquaporin-related genes, such as *aqp5* and *aqp8*, as well as Ca²⁺-activated chloride channel anoctamin 1 (*ano1*), and

antimicrobial peptides-related genes, including *nmb* and *pgq*, in the granular glands were displayed in **c** and **d**, respectively. Several hormone-related genes of granular glands, including *trh*, *xt6l*, and *tph1*, were also evaluated (**e**). The representative results of this analysis were presented in panels **c–e**. *odc* was used as the RNA loading control and **c–e** shared the RNA loading control in **c**. For RT-PCR analysis, each gene was subjected to three repetitions using three replicates of total RNA samples, while each total RNA sample was extracted from the skin of three individual frogs. The original data for the mentioned RT-PCR can be found in Supplementary Fig. 32 and Supplementary Fig. 33. Additionally, the original data for the RT-PCR in Supplementary Fig. 25 can be found in Supplementary Fig. 34. DS, dorsal skin; VS, ventral skin. The black scale bar in **a** is 0.5 mm.

mutation in the *BRAF* oncogene, associated with the development of human nevus and melanoma[57]. This mutation encodes BRAF^V600E, which substitutes valine with glutamic acid at amino acid 600 of the kinase domain, thereby transforming the protein from an inactive to an active form. The resultant BRAF^V600E protein continuously activates the MAPK oncogenic signaling cascade and induces the formation of melanocytic nevus and melanoma in mice and zebrafish when expressed in melanocytes[57]. In this *Xenopus tropicalis* melanoma model, the BRAF^V600E expression was under endogenous *mitf* promoter control in theory. Thus, a similar phenotype of aberrant melanophore proliferation observed in xanthophores and granular glands would indicate the presence of *mitf* expression in xanthophores and granular glands. Mosaic G0 founders of targeted integration of *BRAF^T1799A* into the *mitf* locus, designated as *mitf*-BRAF^V600E, were generated by injecting a mixture of SpCas9 mRNA, gRNA, and donor vector plasmids into zygotes at the 1-cell stage. This resulted in mosaic G0 founders

displaying abnormal proliferations of xanthophores and melanophores, as expected (Fig. 6b). The F1 generation of *mitf*-BRAF^V600E *Xenopus tropicalis* was obtained through outcrossing the mosaic G0 founders with wild-type individuals, as shown in Fig. 6c. The expression of the fusion protein EGFP-BRAF^V600E in Tyr-positive melanophores led to the activation of the MAPK signaling pathway, as demonstrated in Fig. 6d, e. These results indicated that not only did *BRAF^T1799A* integrate correctly as expected, but it also expressed BRAF^V600E under the control of the endogenous *mitf* promoter. Furthermore, the expressed fusion protein exhibited sustained kinase activation function. As expected, melanocytic nevi-like lesions were firstly apparent throughout the bodies of the F1 generations of *mitf*-BRAF^V600E *Xenopus tropicalis*, and corresponding tadpoles at stage 57/58 exhibited abnormal proliferation of xanthophores and melanophores (Fig. 6b, c, f). Adult *mitf*-BRAF^V600E;*cdkn2b*^+/− *Xenopus tropicalis* spontaneously develop nevi caused by abnormal proliferation of melanophores and xanthophoromas caused by

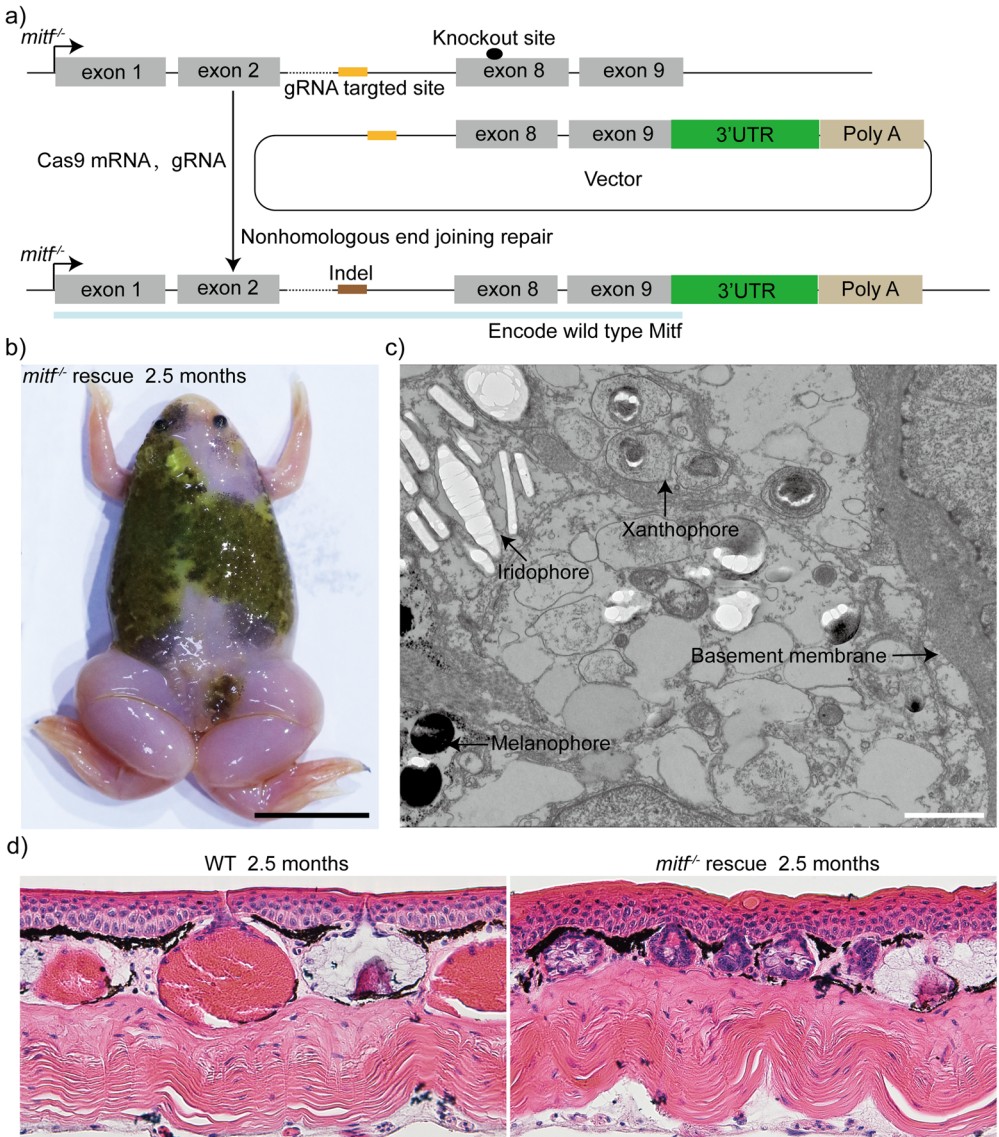

**Fig. 5 | Restoration of the damaged *mitf* site in *mitf*[−/−] *Xenopus tropicalis* results in the rescue of the phenotype of melanophores, xanthophores, and granular glands. a** The precise integration of the DNA fragment encompassing the final two exons and the 3′UTR region of *mitf* into the *mitf*[−/−] genomic locus via CRISPR/Cas9-mediated non-homologous end joining repair has been demonstrated. **b** A rescued phenotype was demonstrated in a representative 2.5-month-old froglet of the mosaic founders. Approximately 300 embryos of *mitf*[−/−] *Xenopus tropicalis* were injected, and eventually, six tadpoles were found to have a rescued phenotype of melaphores. **c** TEM examination of the dorsal skin of the rescued froglet depicted in **b** revealed a significant resurgence of both xanthophores and melanophores. We sampled three distinct regions of the dorsal skin. Six ultrathin sections were subsequently generated from each of the three samples to yield the representative results presented here. **d** The histological structure of the dorsal skin from 2.5-month-old WT froglets and rescued froglets was presented. Three 2.5-month-old WT froglets and three rescued froglets depicted in **b** were subjected to H&E staining, and two dorsal skin samples with 3 mm×3 mm were obtained from each frog. Histological examination was performed on ten paraffin sections for each collected sample to generate the representative results presented here. Scale bar in **b**, 1 cm; Scale bar in **c**, 1 μm; Scale bar in **d**, 50 μm.

abnormal proliferation of xanthophores[58] (Fig. 6g) (A manuscript focusing on the tumor formations, eye degeneration, nevi, and bone dysplasia in this line is currently under preparation). Moreover, after genotyping, we confirmed the correct integration of *BRAF*[T1799A] in the target lines (Fig. 7a and Supplementary Fig. 21). Histopathological assessments revealed abnormal proliferations of melanophores and xanthophores, along with the presence of abundant granular glands in this line (Fig. 7b–d). Consequently, these findings were indicative of the expression of *mitf* in xanthophores and granular glands.

Collectively, the findings reported above demonstrated that abrogating *mitf* via the gRNA T7 resulted in the disappearance of light-absorbing melanophores, granular glands and light-absorbing/reflecting xantho-phores throughout the entirety of *Xenopus tropicalis*' body, revealing that

*mitf* functioned not only as a master regulator of melanophores but also as a key factor in orchestrating the development of xanthophores and granular glands. Still, the exact molecular mechanisms by which *mitf* regulates the development of xanthophores and granular glands remain to be deciphered.

## A colorless and immunodeficient *Xenopus tropicalis* line is established through the *mitf/prkdc/il2rg* triple-knockout

To generate immunodeficient *Xenopus tropicalis* for tumor transplantation, we employed the CRISPR/Cas9 system to simultaneously knock out *prkdc* and *il2rg* genes, which are essential for T cell, B cell, and NK cell development[3]. The dual knockout strategy was depicted in Supplementary Fig. 22a, where SpCas9 mRNA and gRNAs were co-injected into fertilized *Xenopus tropicalis* eggs at the 1-cell stage,

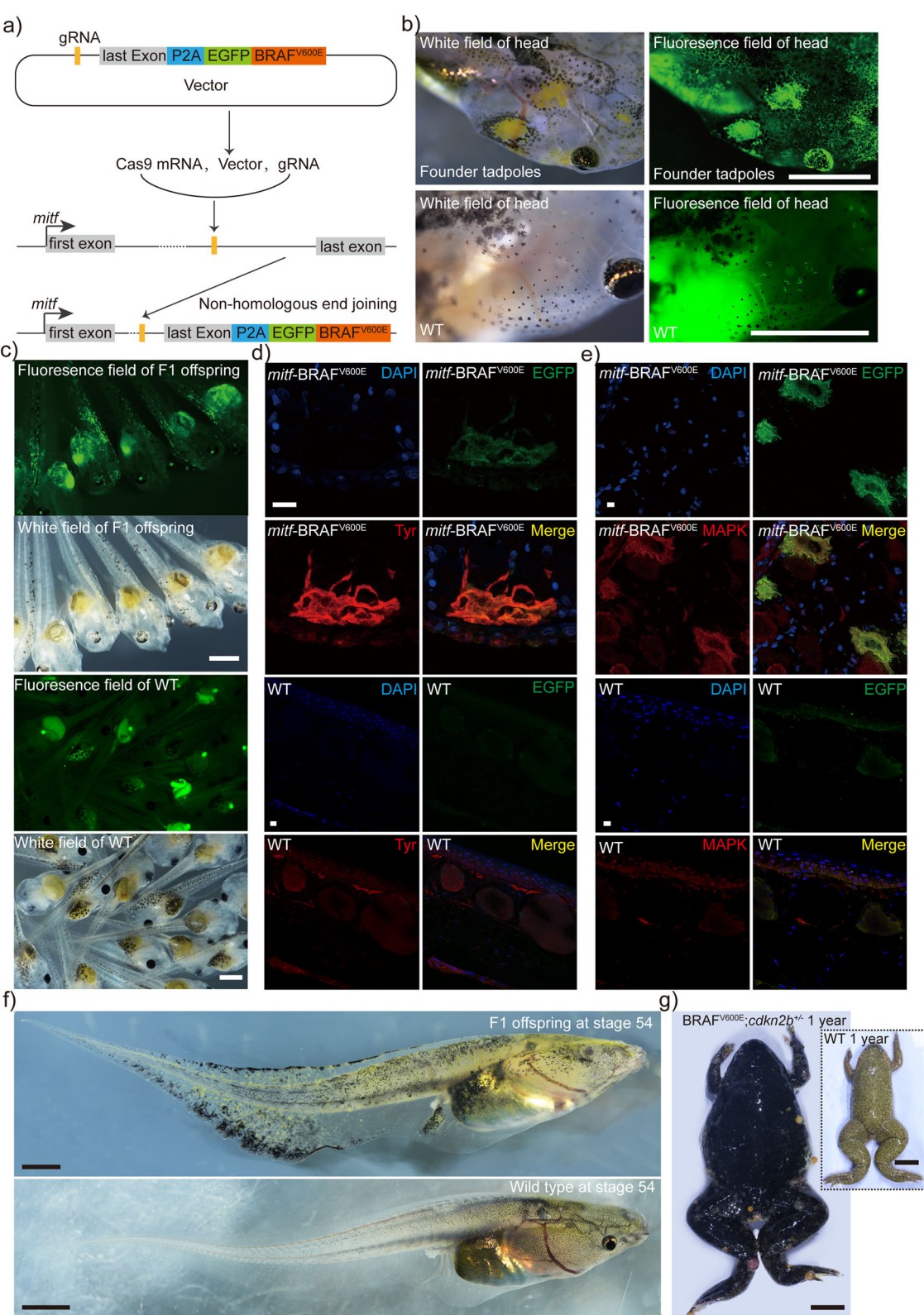

resulting in mosaic G0 founders with a 60% efficiency of knockout for *prkdc* and *il2rg* individually, thereby implying a 36% efficiency for *prkdc/il2rg* double-knockout (Supplementary Fig. 22b). Of these G0 founders, 53% (72/135) exhibited significant thymic abnormalities at stage 56, as shown in Supplementary Fig. 22c. Subsequently, we raised these tadpoles to sexual maturity and carried out assortative mating to produce offspring with various genotypes, selecting

mutants with Indels of *prkdc* and *il2rg* able to cause frameshift mutations. Among these mutants, individuals with 4 bp and 22 bp deletions in *il2rg* alleles, and 11 bp deletions in both *prkdc* alleles, were denoted as *prkdc*⁻/⁻/*il2rg*⁻/⁻ and comprised 20% of the animals (26/128). The thymus and spleen of *prkdc*⁻/⁻/*il2rg*⁻/⁻ *Xenopus tropicalis* were stunted and structurally abnormal, with a significant downregulation of T-cell, B-cell, and NK-cell marker genes in the

**Fig. 6 | The integration of the *BRAF*^T1799A within the *mitf* locus elicits spontaneously formed melanophores-related nevi and xanthophores-like moles in *Xenopus tropicalis*.** **a** The strategy of precisely integrating the human *BRAF*^T1799A mutation into the *mitf* genomic locus through non-homologous end joining repair mediated by CRISPR/Cas9 has been demonstrated. This involves placing the mutation downstream of the last exon of *mitf*, thereby utilizing the endogenous promoter of *mitf* to drive BRAF^V600E expression. **b** At stage 56, following precise integration of human *BRAF*^T1799A into the *mitf* genomic locus, the founder generation tadpoles exhibited a mosaic phenotype characterized by aberrant proliferation of melanophores and xanthophores. Additionally, a significant proliferation of xanthophores was observed using spontaneous green fluorescence. The control group exhibited the absence of this phenotype in WT tadpoles. A total of eight such tadpoles were observed at this stage. **c** Compared to the control group of WT tadpoles, the F1 generation *mitf*-BRAF^V600E tadpoles at stage 46 were observed and

found to exhibit EGFP signals. **d, e** Immunofluorescence analysis was performed on three tail samples obtained from three stage 56 *mitf*-BRAF^V600E tadpoles from the F1 generation. Every sample was fixed, embedded, and sectioned into six paraffin slices for the analysis. The results demonstrated co-localization of Tyr and EGFP (**d**), as well as co-localization of EGFP and MAPK signaling pathway activation (**e**). The WT control group showed an absence of EGFP signals, and cells with activated MAPK pathway were also scarce. The activation of the MAPK signaling pathway was evidenced by positive fluorescence signal of the antibody-ERK1 + ERK2. The images presented here were representative of the observed results. **f** A typical photograph of F1 *mitf*-BRAF^V600E tadpoles at stage 54 and the corresponding batch of WT tadpoles was presented. **g** Compared to a 1-year-old WT frog, a representative image of a 1-year-old *cdkn2b*^+/−;*mitf*-BRAF^V600E frog showed spontaneously formed nevi and xanthophoromas. Scale bar in **b**, 2 mm; Scale bar in **c**, 1 mm; Scale bar in **d** and **e**, 10 µm; Scale bar in **f**, 5 mm; Scale bar in **g**, 1 cm.

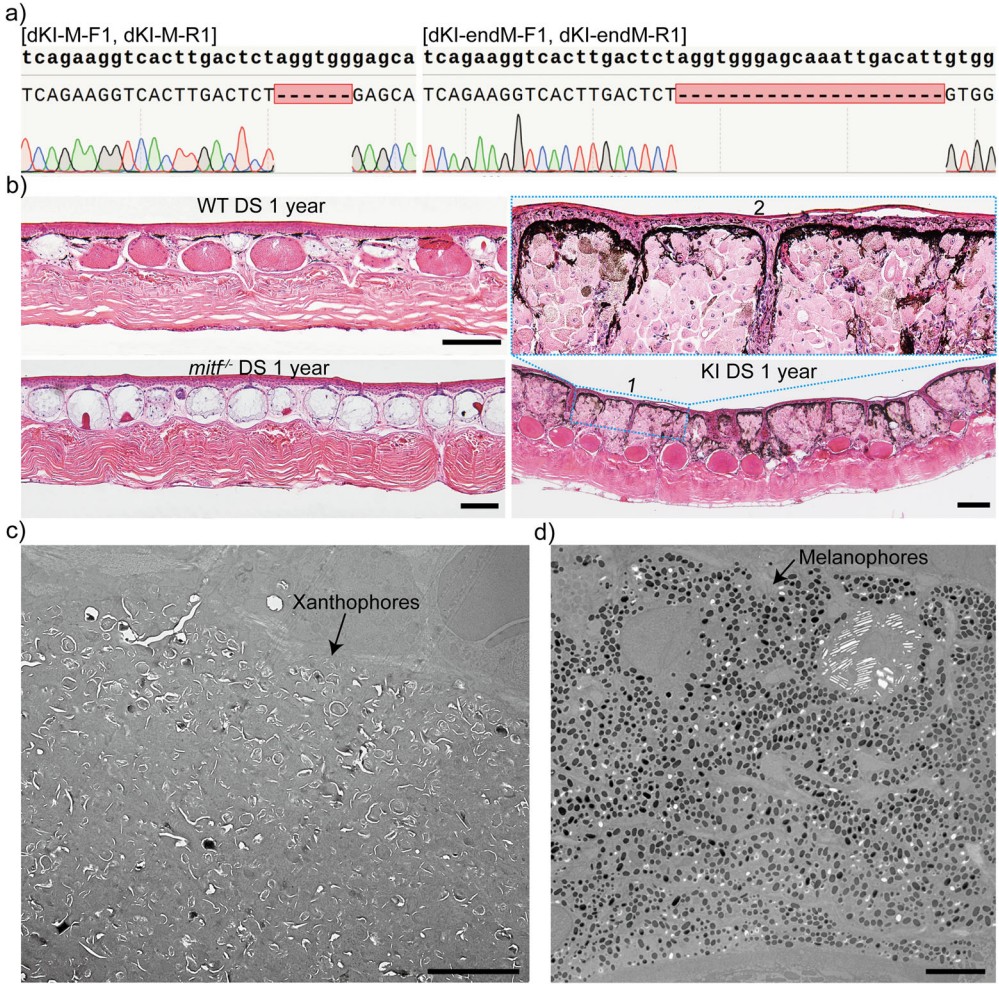

**Fig. 7 | The integration of the *BRAF*^T1799A within the *mitf* locus results in a substantial proliferation of melanophores and xanthophores, as well as a significant increase in the number of granular glands.** **a** The Sanger sequencing of the integrated upstream and downstream repair site confirmed the correct integration of *BRAF*^T1799A within the design target. For a detailed explanation of the detection principle, please refer to Supplementary Fig. 18. Ten *mitf*-BRAF^V600E tadpoles were subjected to genotyping. **b** The histological structure of the dorsal skin (DS) from 1-year-old WT, F2 *mitf*^−/−, and F1 *mitf*-BRAF^V600E *Xenopus tropicalis* specimens was depicted. Blue dashed box 2 corresponded to the magnified view of blue dashed box 1. Five frogs of each

genotype underwent H&E staining. Two dorsal skin samples were collected from each frog. Histological examination of each collected sample was conducted on 10 paraffin sections for generating the representative results presented here. **c, d** Upon TEM examination of the dorsal skin of 1-year-old F1 *mitf*-BRAF^V600E *Xenopus tropicalis*, a notable proliferation of xanthophores (**c**) and melanophores (**d**) was observed. We conducted TEM analysis on three frogs of *mitf*-BRAF^V600E *Xenopus tropicalis*, collecting three samples from distinct regions on the dorsal skin for each frog. Six ultrathin sections were generated from each sample to generate the representative results presented here. The black scale bar in **b** is 100 µm. The scale bar in **c, d** is 5 µm.

spleen, liver, lung, and blood, indicating a deficiency in the development of T cells, B cells, and NK cells (Supplementary Fig. 23 and Supplementary Fig. 24). The significant reduction in *il2* expression in the spleen of *prkdc*^−/−/*il2rg*^−/− *Xenopus tropicalis* further indicates abnormal T-cell development in these frogs (Supplementary Fig. 25).

Additionally, we performed skin transplantation using *Xenopus tropicalis* of the wild type and *prkdc*^−/−/*il2rg*^−/− mutants as an indicator of immune deficiency[59]. The results showed that wild type *Xenopus tropicalis* exhibited immunological rejection to allogeneic or xenogeneic skin grafts (The skin transplants from *Xenopus tropicalis*,

*Xenopus laevis*, and zebrafish, all exhibited a certain immune rejection response with a 3/3 rate), while the *prkdc$^{-/-}$/il2rg$^{-/-}$* mutants accepted skin grafts from different species with varying degrees of rejection depending on evolutionary distance (3/3 *Xenopus tropicalis* skin transplants showed no rejection after one year, while 3/3 *Xenopus laevis* had mild rejection, and 3/3 zebrafish exhibited strong rejection after 14 days), for instance, completely accepting skin grafts from allogeneic *Xenopus tropicalis* yet showing some immunological rejection to skin grafts from xenogeneic zebrafish (Fig. 8a and Supplementary Fig. 26). Overall, these findings indicated that *prkdc$^{-/-}$/il2rg$^{-/-}$ Xenopus tropicalis* mutants were promising subjects for tumor cell transplantation studies across species while indicating their potential utility in tumor immunology research.

Next, we investigated whether *Xenopus tropicalis* with a triple-knockout of *mitf/prkdc/il2rg* was both colorless and immunodeficient. To generate this line, *mitf$^{-/-}$* mutants were crossbred with *prkdc$^{-/-}$/il2rg$^{-/-}$* individuals, resulting in F2 generations that showed colorlessness and the *mitf/prkdc/il2rg* triple-knockout genotype (*mitf$^{-/-}$/prkdc$^{-/-}$/il2rg$^{-/-}$*) in a frequency of 22/1086, following Mendelian inheritance principles (Supplementary Fig. 27). To assess the immunodeficiency of this strain, we performed allotransplantation of wild-type skin to both *mitf$^{-/-}$* and *mitf$^{-/-}$/prkdc$^{-/-}$/il2rg$^{-/-}$ Xenopus tropicalis*, and observed immune rejection responses and T cell infiltration. Skin allograft transplantation experiments conducted in thymectomized and wild-type *Xenopus laevis* have provided evidence that CD4 and CD8 positive T lymphocytes play a crucial role in the rejection response of skin grafts in *Xenopus* frogs[59]. Results showed that recipients of *mitf$^{-/-}$* individuals experienced immune rejection with an elevated number of T cells (CD3$^+$ cells) in the transplanted skin, whereas recipients of *mitf$^{-/-}$/prkdc$^{-/-}$/il2rg$^{-/-}$* individuals did not experience immune rejection and the transplanted skin developed similarly to skin from wild type frogs without transplantation (Fig. 8b–e). Thus, these results demonstrated that *mitf$^{-/-}$/prkdc$^{-/-}$/il2rg$^{-/-}$ Xenopus tropicalis* was a colorless and immunodeficient line.

### Visualizing allografted tumor cell metastasis in *mitf$^{-/-}$/prkdc$^{-/-}$/il2rg$^{-/-}$ Xenopus tropicalis*

To ascertain the suitability of *mitf$^{-/-}$/prkdc$^{-/-}$/il2rg$^{-/-}$ Xenopus tropicalis* for tumor allotransplantation, we conducted allotransplantations of xanthophoromas and melanomas in these amphibians. The xanthophoroma tissues used for allotransplantation were sourced from the skin of *cdkn2b$^{-/-}$;mitf*-BRAF$^{V600E}$ *Xenopus tropicalis*. When these donors of xanthophoroma tissues were transplanted to wild type and *mitf$^{-/-}$/prkdc$^{-/-}$/il2rg$^{-/-}$ Xenopus tropicalis* recipients, no visible metastasis was observed within a month for both recipients (Supplementary Fig. 28a–e). Xanthophoromas metastasis was observed in the *mitf$^{-/-}$/prkdc$^{-/-}$/il2rg$^{-/-}$* recipients for up to two months post-transplantation, while no signs of metastasis were seen in the wild type recipients during the same period (Supplementary Fig. 28f–g). At five months post-transplantation, systemic metastasis of xanthophoromas was evident in the *mitf$^{-/-}$/prkdc$^{-/-}$/il2rg$^{-/-}$* recipients, but not in the wild type recipients (Fig. 9a). Furthermore, through a comparative analysis of the green fluorescence signal in the dorsal thigh skin of the transplanted *mitf$^{-/-}$/prkdc$^{-/-}$/il2rg$^{-/-}$* recipient with that of the *mitf$^{-/-}$* frog, we discovered that *mitf$^{-/-}$/prkdc$^{-/-}$/il2rg$^{-/-}$ Xenopus tropicalis* was suitable for intravital imaging of dynamic changes in tumor cells using green fluorescence signals post-transplantation (Fig. 9b). Conversely, wild type *Xenopus tropicalis* was deemed unsuitable due to the presence of intense autofluorescence. Remarkably, xanthophoromas were observed to metastasize to the entire body of *mitf$^{-/-}$/prkdc$^{-/-}$/il2rg$^{-/-}$* recipients about one year after transplantation, resulting in grave lesions necessitating the sacrifice of the recipients (Supplementary Fig. 28h). Conversely, the transplanted xanthophoromas in wild type recipients stayed at the site of transplantation, as opposed to metastasizing or being eliminated by the host immune system (Supplementary Fig. 28i). These findings suggested

that the use of *mitf$^{-/-}$/prkdc$^{-/-}$/il2rg$^{-/-}$ Xenopus tropicalis* was a viable method for tumor allogeneic transplantation and intravital imaging of transplanted tumor cells. Furthermore, the observed mutual restriction mechanism between the wild type recipients and transplanted xanthophoromas may have implications for the development of tumor immunotherapy.

Melanoma tissues from *tp53$^{Δ7/Δ7}$ Xenopus tropicalis* donors were utilized for melanoma allotransplantations[60]. As previously reported, the melanoma developed similarly to human melanoma[60]. Due to limited nevi and melanoma samples, four dysplastic nevi and four metastatic melanomas were selected and transplanted into *mitf$^{-/-}$/prkdc$^{-/-}$/il2rg$^{-/-}$* recipients (experimental group) and *mitf$^{-/-}$* recipients (control group), respectively, as demonstrated in Supplementary Fig. 29a. Unfortunately, one dysplastic nevus and one metastatic melanoma that were transplanted into their respective groups became detached accidentally on the fourth day after transplantation, leaving only one transplanted frog per group to continue with the experiment. Thirty days post-transplantation of dysplastic nevi, migration of donor melanophores was observed exclusively in the recipient skin of *mitf$^{-/-}$/prkdc$^{-/-}$/il2rg$^{-/-}$ Xenopus tropicalis*, but not in that of the *mitf$^{-/-}$* frog (Supplementary Fig. 29b–g). Notably, even after 40 days of transplantation in the skin of the *mitf$^{-/-}$* frog, the migration of donor melanophores remained undetectable (Fig. 10a, c). However, the migration of donor melanophores was observed in the *mitf$^{-/-}$/prkdc$^{-/-}$/il2rg$^{-/-}$* recipient within the same time frame, and the morphology of these donor melanophores was indistinguishable from that of differentiated *Xenopus tropicalis* melanophores[12] (Fig. 10b, d). As for metastatic melanoma tissues allotransplantations, migration of melanoma cells was also exclusively observed in the *mitf$^{-/-}$/prkdc$^{-/-}$/il2rg$^{-/-}$* recipient (Supplementary Fig. 23j–m and Fig. 10e–h). Unlike dysplastic nevi transplantation results, the morphology of migrating melanomas exhibited an undifferentiated state after 40 days of transplantation (Fig. 10e–h). To supplement the inventory of melanoma transplants, allogeneic transplantation experiments were conducted utilizing spontaneously occurring dysplastic nevi from *cdkn2b$^{-/-}$/tp53$^{-/-}$ Xenopus tropicalis*. The outcomes revealed that *mitf$^{-/-}$* recipients manifested immune rejection, contrasting with the absence of such rejection in *mitf$^{-/-}$/prkdc$^{-/-}$/il2rg$^{-/-}$* recipients (Supplementary Fig. 30). Dysplastic nevi transplanted onto both *mitf$^{-/-}$* and *mitf$^{-/-}$/prkdc$^{-/-}$/il2rg$^{-/-}$* recipients did not exhibit notable melanophore migration, potentially attributed to the advanced age of the recipients. Taken together, the transplantation of dysplastic nevi and metastatic melanomas in *mitf$^{-/-}$ Xenopus tropicalis* recipients was observed to result in a lack of migration and clear signs of clearance by the host's immune system (Fig. 10d, h). Interestingly, xanthophoromas and melanomas transplanted into immunocyte-normal recipients (wild type and *mitf$^{-/-}$ Xenopus tropicalis*) demonstrated contrastive behaviors: persistent in situ growth and immune rejection, respectively. These discrepancies could be attributed to either variable oncogenic potential of donor tumors or idiosyncratic interactions between tumorous cells and the recipient's immune system.

Collectively, as shown in Fig. 11, our studies suggested that *mitf$^{-/-}$/prkdc$^{-/-}$/il2rg$^{-/-}$ Xenopus tropicalis* was a valuable tool for transplanting tumors and intravitally observing the initial stages of metastasis, particularly the migration of melanomas. Furthermore, our results also implied that a comprehensive examination of tumor response following transplantation of xanthophoromas and melanomas in *Xenopus tropicalis* could hold promise for advancing the development of tumor immunotherapy.

### Discussion

In this study, we developed an animal model of tumor allograft recipient utilizing a genetically engineered amphibian, the colorless and immunodeficient *Xenopus tropicalis* line generated by *mitf/prkdc/il2rg* triple-knockout, enabling intravital observations of transplanted xanthophoromas' and melanomas' migration and metastasis. Concomitantly, we discovered that *mitf$^{-/-}$ Xenopus tropicalis* lacked not only melanophores but

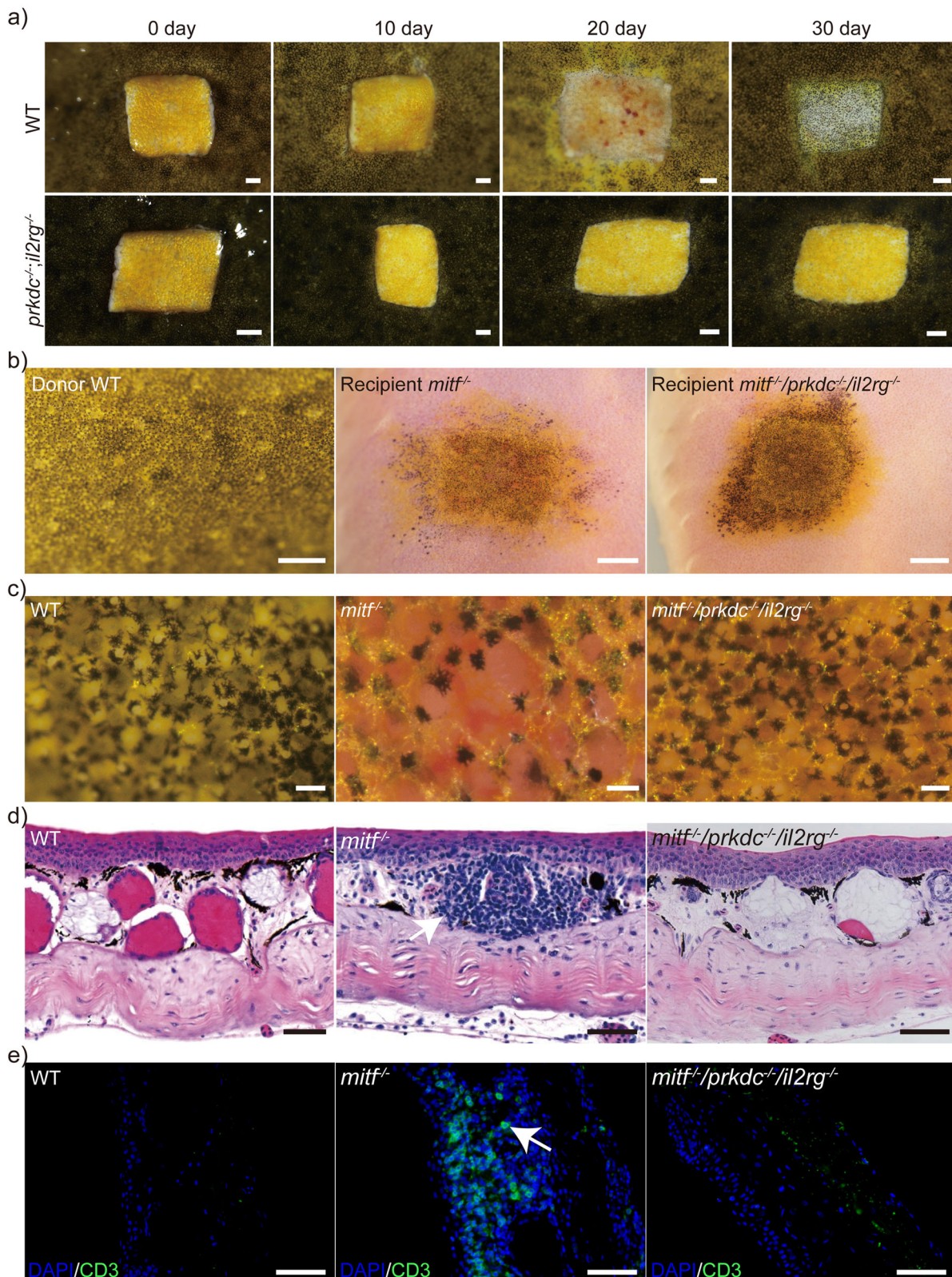

also xanthophores and granular glands. By integrating *BRAF*[T1799A] into the *mitf* locus and conducting rescue experiments of *mitf*[−/−] damage site targeted repair, we demonstrated that *mitf* was also expressed in xanthophores and granular glands, and played a crucial role in their development, revealing novel functions for the *mitf* locus in the development of xanthophores and granular glands.

In melanocyte development, several signaling pathways, including MC1R, PI3K, MAPK, and WNT, intricately regulate the process through the mediation of MITF[26,61]. MITF's regulatory influence manifests through the modulation of its expression levels. For instance, the collaborative orchestration of the PI3K, MAPK, and WNT pathways regulates MITF expression by manipulating the transcription factor BRN2 (POU3F2), thus

**Fig. 8 | The *mitf/prkdc/il2rg* triple-knockout results in the creation of a line of colorless and immunodeficient *Xenopus tropicalis*. a** Three WT and three *prkdc⁻/⁻/il2rg⁻/⁻ Xenopus tropicalis* aged 6 months were used as transplant recipients. The donors were three *tyr⁻/⁻ Xenopus tropicalis* aged 2 years, and their dorsal skins were grafted onto the recipients' dorsal skin. The grafting outcomes were monitored for a 30-day period post-transplantation. The representative results were presented here and the transplanted donor skins were approximately 5 mm × 5 mm in size. **b–e** Three *mitf⁻/⁻* and three *mitf⁻/⁻/prkdc⁻/⁻/il2rg⁻/⁻ Xenopus tropicalis* froglets were selected as transplant recipients at the age of two months. The donor skins were obtained from three WT *Xenopus tropicalis* at the same age and were grafted onto the dorsal skin of the recipients. The transplanted donor skins were approximately 3 mm×3 mm in size, and the WT skin and representative photos of the transplanted donor skins were taken on the 52nd day post-transplantation (**b**). The magnified view of the corresponding skin in **b** was shown in **c**. **d** The histological structure of the corresponding skin in **c** was presented through histological examination of 18 paraffin sections from the collected sample, which provided the representative results presented here. **e** T cell infiltration following skin allografts in *mitf⁻/⁻* and *mitf⁻/⁻/prkdc⁻/⁻/il2rg⁻/⁻ Xenopus tropicalis*. T cells were labeled with the CD3 antibody, and cell nuclei were stained with DAPI. The sample preparation method was the same as for **d**. Scale bar in **a** and **b**, 1 mm; Scale bar in **c**, 100 μm; Scale bar in **d** and **e**, 50 μm.

influencing MITF's function[26]. Additionally, MITF's functionality is also subject to control through the regulation of its phosphorylation status. The phosphorylation at S73, S69, S298, and the three C-terminal sites are modulated by PI3K, MAPK, and WNT signaling pathways, impacting MITF's activity[62–64]. Research on the regulation of Mitf in melanocyte development in *Xenopus tropicalis* is limited. Nevertheless, considering the conservation of melanocyte development in vertebrates, the regulation of Mitf protein stability, nuclear translocation, and protein expression levels plays a crucial role in controlling melanocyte development. While the role of *Mitf* in regulating melanocytes and melanophores is well-established, its involvement in xanthophores has not been extensively studied[26]. Mutated zebrafish carrying *mitfa^{C417T/C417T}*, which produces a stop codon substituting glutamine 113, display an increase in iridophores but an absence in melanophores and a slight reduction in xanthophore pigmentation[20]. Knockout of *mitf* in tilapia resulted in the loss of melanophores, increased iridophores and erythrophores, and enlarged xanthophores[65]. The underlying mechanisms behind the altered xanthophores in zebrafish and tilapia are still unclear, and knockout of *mitf* in fish leading to the loss of xanthophores has not been reported. In zebrafish, xanthophores and melanophores can be derived from a common Tuba8l3-positive progenitor cell, and the developmental fate of these progenitor cells is determined by the expression of *mitfa* and *kita* for melanophores while *csf1ra* for xanthophores[51]. Recent studies using single-cell RNA sequencing in zebrafish have confirmed that *mitfa* expression occurs in xanthophore progenitor cells[27]. Additionally, in vitro studies have shown that amphibian xanthophores could transdifferentiate into melanophores[66]. It is thus plausible that *Xenopus tropicalis* utilizes a similar mechanism wherein both melanophores and xanthophores stem from a common progenitor cell, and *mitf* expression may be crucial for the development of these cells. A potential consequence of *mitf* loss-of-function in *Xenopus tropicalis* would therefore be the absence of both xanthophores and melanophores.

The granular glands of *Xenopus tropicalis* exhibit strong similarities in cellular composition and gene expression in comparison to the exocrine sweat glands of humans and mice[45–48,67]. Nevertheless, no studies have been conducted on the role of MITF in sweat gland development in either of these species[46]. Here, our study demonstrated the crucial role of Mitf in the development of granular glands in *Xenopus tropicalis* through evidence supporting the genotype-phenotype specificity of granular gland loss caused by *mitf* knockout. While *Mitf* mutant studies in mice primarily focused on melanocyte-related diseases, the potential sweat gland-related phenotype has not been reported, possibly because sweat glands are relatively small and located in the palmar-plantar region[26,46]. Given the complexity of the transcriptional regulatory network of the *mitf* locus[26,68], it was possible that the granular glands specific transcript of *mitf* may lose its function due to the disrupted site in *mitf⁻/⁻ Xenopus tropicalis*, leading to a loss of granular glands in these frogs. However, specific *mitf* transcripts in the oocytes and RPE of *mitf⁻/⁻ Xenopus tropicalis* may remain unaffected by the mutants, thus allowing for normal melanin deposition in oocytes and RPE development in eyes (Supplementary Fig. 31), similar to some *Mitf/mitf* mutated mice and zebrafish, which exhibit normal RPE-specific *mitf* transcripts[26,68]. Therefore, exploring the molecular mechanism underlying granular gland loss in *mitf⁻/⁻ Xenopus tropicalis* represents an interesting issue to address, and a thorough examination of *Mitf* functions in mouse sweat glands is warranted.

In humans, the MITF locus also plays a crucial role in in the development and function of various immunocytes, such as dendritic cells, B cells, and mast cells[69]. The *mitf* locus in *Xenopus tropicalis* is hypothesized to share similarities with the *Mitf* locus in mammals, allowing for the transcription of multiple isoforms, each exhibiting distinct expression patterns and functions. In mammals, it is well-established that *Mitf* is also expressed in mast cells, and various mast cell defects have been observed in several *Mitf* mutant mouse strains[69]. *Mitf* plays a pivotal role in guiding the differentiation of pre-BMPs (pre-basophil and mast cell progenitors) into mast cells[69]. Given the essential role of mast cells in the inflammatory process, the impact of *mitf* knockout on immune rejection defects in *Xenopus tropicalis* is anticipated to be relatively modest. This perspective is reinforced by the notable infiltration of CD3⁺ cells in *mitf⁻/⁻ Xenopus tropicalis* following allogeneic skin transplantation (Fig. 8). Additionally, the loss of granular glands in *mitf⁻/⁻ Xenopus tropicalis* results in a weakened immune response due to the absence of antibacterial substances secreted by these glands. Despite this, allograft transplantation in *mitf⁻/⁻* recipients results in an increased number of immunocytes and subsequent immune rejection, ultimately leading to allograft destruction. Thus, in the colorless and immunodeficient *Xenopus tropicalis* line of *mitf/prkdc/il2rg* triple-knockout, further exploration is needed to determine the contribution of the *mitf* knockout to the immune deficiency of this organism.

In conclusion, the *mitf⁻/⁻/prkdc⁻/⁻/il2rg⁻/⁻ Xenopus tropicalis* provide a valuable recipient model for studying the molecular mechanisms of tumor metastasis in this species and a possibility for establishing xenograft models in amphibians in the future.

## Methods

### Xenopus tropicalis and Xenopus laevis maintenance

Adult *Xenopus tropicalis* and *Xenopus laevis* were procured from Nasco (Fort Atkinson, WI, USA; http://www.enasco.com) and were maintained at the laboratory facility in compliance with approved standards for husbandry and management. Specifically, for *mitf⁻/⁻/prkdc⁻/⁻/il2rg⁻/⁻* or *prkdc⁻/⁻/il2rg⁻/⁻ Xenopus tropicalis*, despite being maintained under the same husbandry conditions as WT *Xenopus tropicalis*, it is essential to keep the rearing water clean and promptly remove food residues and excreta. All the experimental procedures involving *Xenopus tropicalis* and *Xenopus laevis* were conducted under the approval of the Institutional Animal Care and Use Committee at the Southern University of Science and Technology. We have complied with all relevant ethical regulations for animal use.

### Polymerase chain reaction (PCR), reverse transcription-polymerase chain reaction (RT-PCR) analysis, and quantitative real-time PCR (qPCR)

To conduct PCR, we employed *TransTaq*® DNA Polymerase High Fidelity (HiFi) (AP131-11, Transgen Biotech) as the manual. To perform RT-PCR analyses, we employed the TransZol Up Plus RNA Kit (ER501-01, Transgen Biotech) to extract total RNA, followed by cDNA synthesis using the *TransScript*® Uni One-Step gDNA Removal and cDNA Synthesis SuperMix (AU311-03, Transgen Biotech). The *TransTaq*® DNA Polymerase High Fidelity (HiFi) (AP131-11, Transgen Biotech) was also used in RT-PCR and qPCR analyses. The specific PCR, RT-PCR and qPCR primer sequences utilized in this study were listed in Supplementary Data 3.

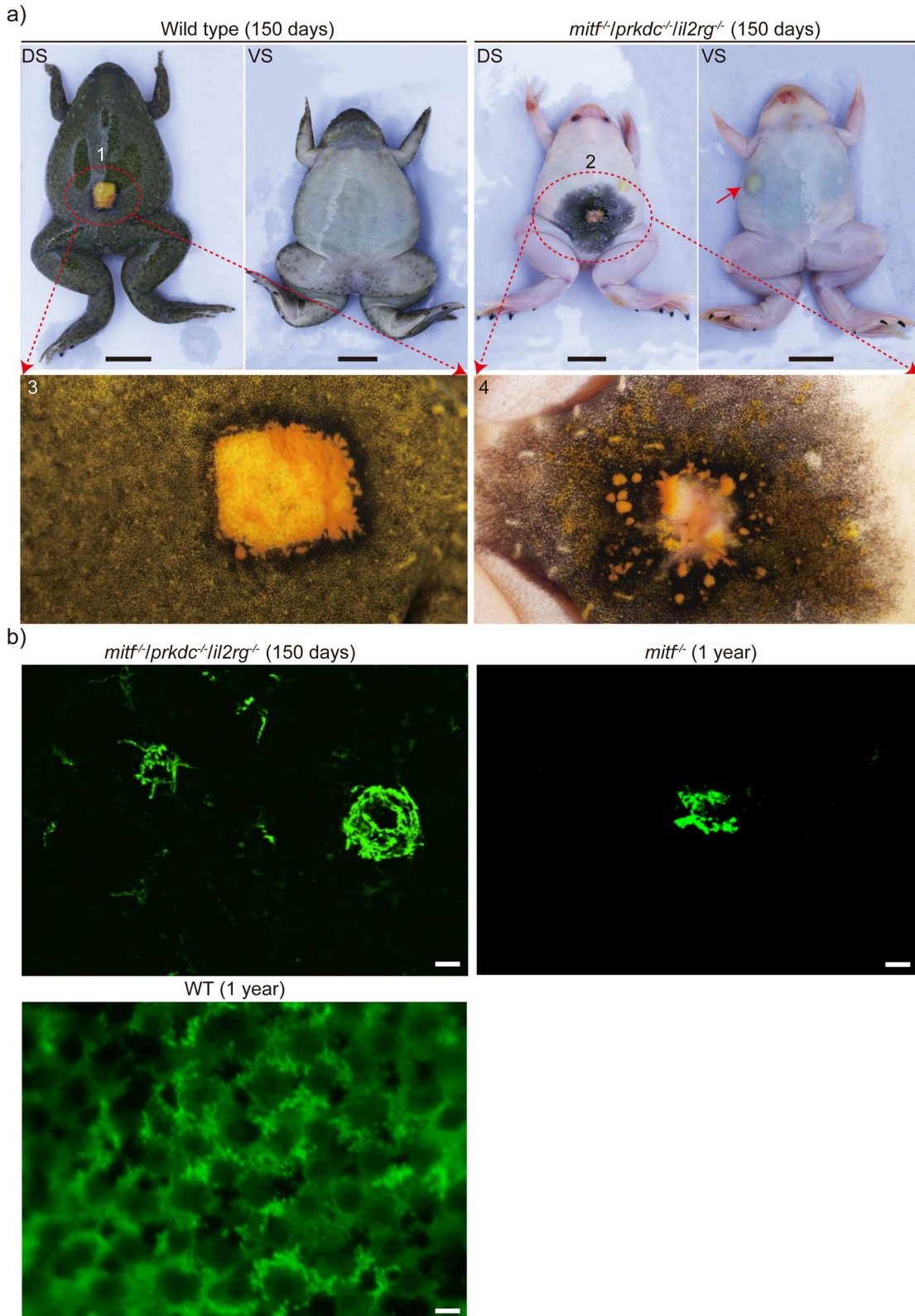

## Preparation of Cas9 mRNA, guide RNA (gRNA) and vectors, embryo microinjection, genotyping and off-target analysis

Preparations of Cas9 mRNA and guide RNA were performed according to the procedure in our previously published methodology[60]. Specifically, the linearization of pCS2-SpCas9 plasmid was facilitated via the *Not I* endonuclease and ensued in vitro transcription through the mMessage mMachine SP6 Kit (Ambion, Austin, TX, USA). The gRNA target sequences for *mitf*, *prkdc*, and *il2rg* were compiled in Supplementary Data 3. The gRNA template was obtained by PCR amplification utilizing the pUC57-T7-gRNA scaffold vector, along with the Trac-reverse primer and a gRNA forward primer harboring a T7 promoter and the specific gRNA target sequence. The primer sequences were listed in Supplementary Data 3.

**Fig. 9 | Allogeneic transplantations of xanthophoromas were conducted in** *mitf*$^{-/-}$/*prkdc*$^{-/-}$/*il2rg*$^{-/-}$ *Xenopus tropicalis*, **resulting in the observed systemic metastasis of transplanted xanthophoromas in the recipient animals. a** At 150 days post-transplantation, xanthophoromas were observed intravitally in both dorsal skin (DS) and ventral skin (VS), and the enlarged views of transplanted xanthophoromas in ellipses 1 and 2 were depicted in 3 and 4, respectively. Xanthophoromas from eight stage 58 *cdkn2b*$^{-/-}$;*mitf*-BRAF$^{V600E}$ tadpoles were transplanted onto the dorsal skin of four WT frogs and four *mitf*$^{-/-}$/*prkdc*$^{-/-}$/*il2rg*$^{-/-}$ frogs, which served as the transplant recipients. The transplants were monitored for up to one year. The presented data represented representative results on day 150

post-transplantation, and additional representative observation data are provided in Supplementary Fig. 28. **b** Metastasized xanthophoroma cells were intravitally observed under fluorescence field conditions in the right thigh of the recipient depicted in panel **a**. To provide a representative control, intravital observations were also conducted on the right thigh of three 1-year-old *mitf*$^{-/-}$ frogs that had not undergone xanthophoroma transplantation, using fluorescence field conditions. Control group WT exhibits spontaneous green fluorescence on the dorsal skin of one-year-old *Xenopus tropicalis*. The scale bar in **a** is 1 cm. The scale bar in **b** is 10 μm.

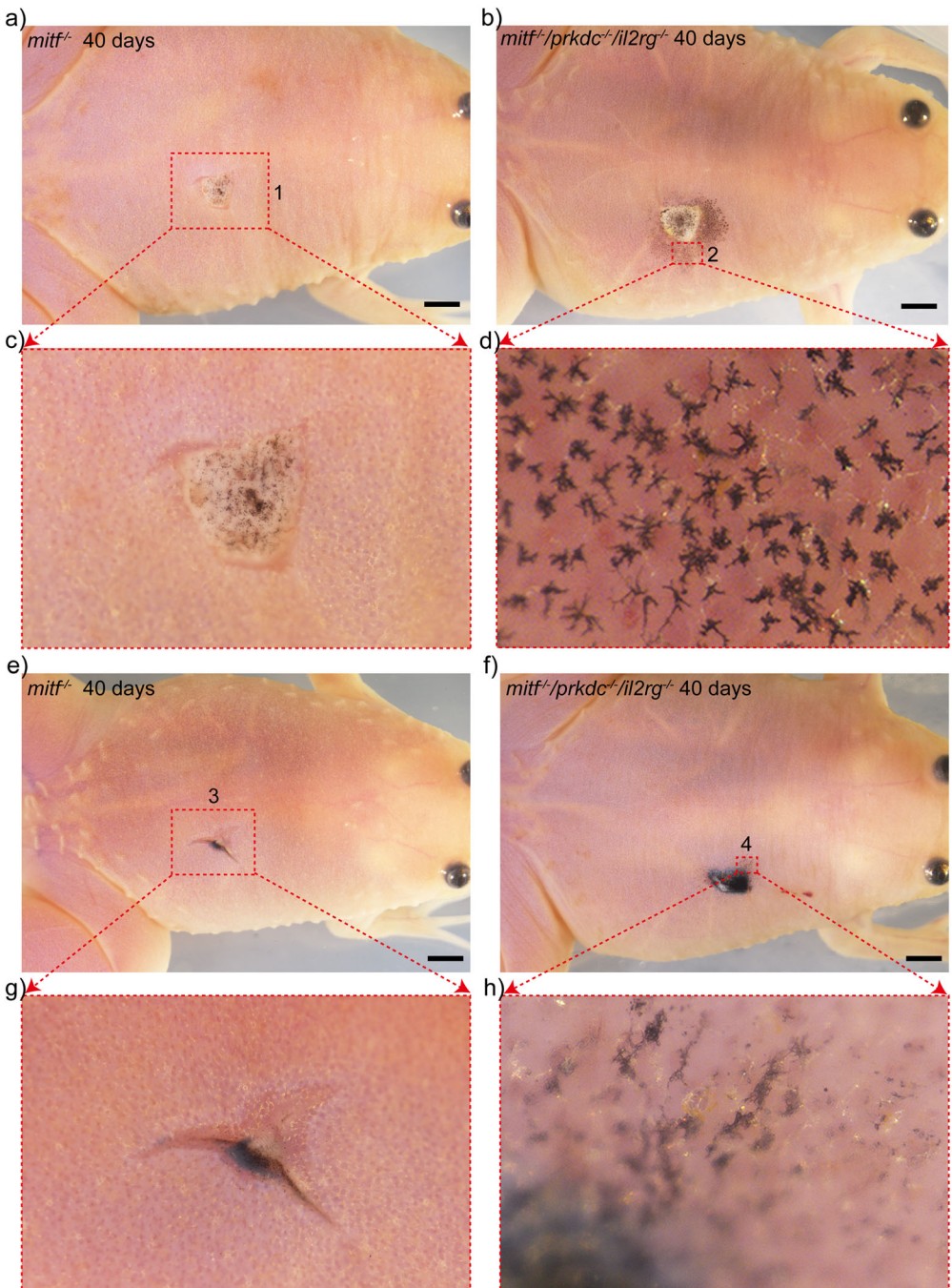

**Fig. 10 | Allogeneic transplantations of melanomas were conducted in** *mitf*$^{-/-}$/*prkdc*$^{-/-}$/*il2rg*$^{-/-}$ *Xenopus tropicalis*, **resulting in the observed migrations of transplanted melanomas in the recipient animals. a–h** At day 40 post-transplantation, the transplanted nevi and metastatic melanoma were examined in *mitf*$^{-/-}$

(**a, c, e, g**) and *mitf*$^{-/-}$/*prkdc*$^{-/-}$/*il2rg*$^{-/-}$ (**b, d, f, h**) recipients. The enlarged views of the transplanted nevi and metastatic melanoma were portrayed in panels **c** and **d** (red dashed boxes 1 and 2) and **g** and **h** (red dashed boxes 3 and 4), respectively. Each scale bar is 1 mm.

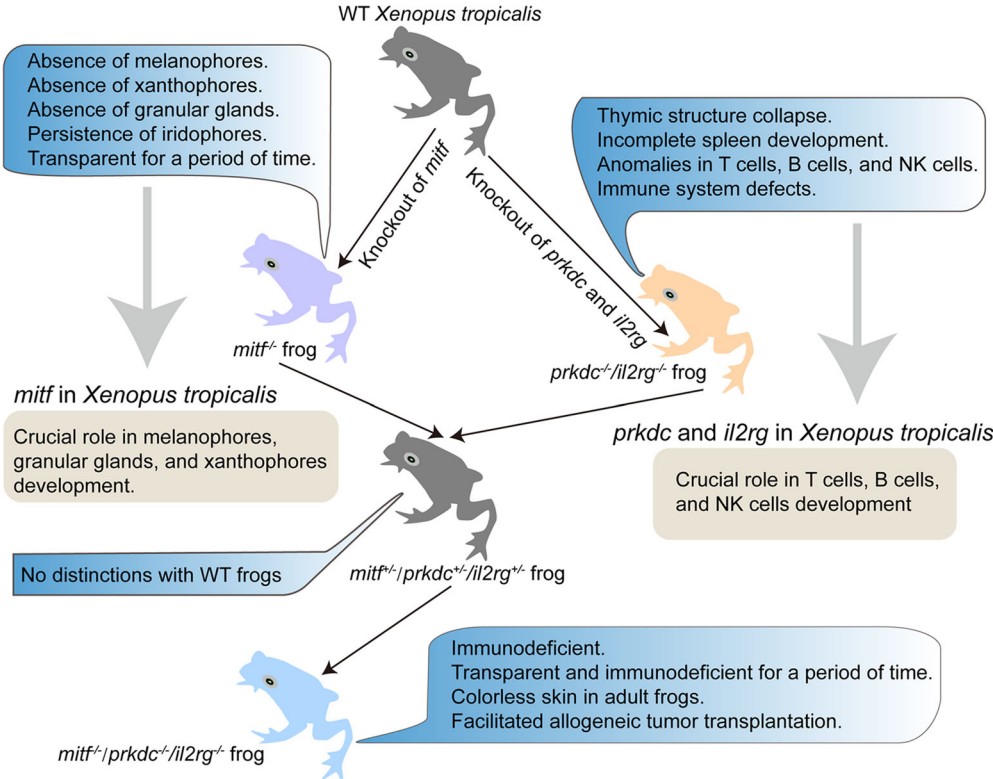

**Fig. 11 | Schematic of *mitf⁻/⁻/prkdc⁻/⁻/il2rg⁻/⁻ Xenopus tropicalis* construction.** The process revealed the crucial role of *mitf* in the development of melanophores, xanthophores, and granular glands. The resulting *mitf⁻/⁻/prkdc⁻/⁻/il2rg⁻/⁻ Xenopus tropicalis* is suitable for research on allograft tumor transplantation.

The gRNA in vitro transcription was conducted using the Transcript Aid T7 High Yield Transcription Kit (Thermo Fisher Scientific, Rockford, IL, USA). For purification of Cas9 mRNA and gRNA, the RNeasy Mini Kit and miRNeasy Mini Kit (Qiagen, Hilden, Germany) were employed, respectively.

To create the pBS-*mitf.le*-P2A-Cre^ERT2-P2A-EGFP-BRAF^T1799A-SV40 poly(A)-*ela*-GFP-SV40 poly(A) donor vector for the *mitf* locus, a *Xenopus tropicalis* genomic fragment (named mitf.le) containing the target site and last exon was amplified by PCR and inserted into the pBluescript II SK⁺ plasmid (Agilent Technologies, Santa Clara, CA, USA). *BRAF*^T1799A was obtained through homologous recombination with a PCR product generated from a PCR-amplified human genomic fragment from Hela Cells, which was a kind gift from Chunhui Hou Lab (Kunming Institute of Zoology, Chinese Academy of Sciences, Kunming, China). The *pEASY*®-Basic Seamless Cloning and Assembly Kit (CU201-02, Transgen Biotech) was used for the homologous recombination. A *Not I*-digested *ela*-GFP fragment from pElastase-GFP, containing a 203 bp rat pancreas-specific Elastase I promoter, GFP coding sequence, and an SV40 polyadenylation signal, was inserted as a reporter. A pBS-*mitf.le(8-9)*-3′UTR donor vector for repairing the *mitf* disrupted site of *mitf⁻/⁻ Xenopus tropicalis* was also created by cloning a PCR-amplified *Xenopus tropicalis* genomic fragment (named mitf.le(8-9)) containing the integration target site and the last two exons into the pBluescript II SK⁺ plasmid. The FastDigest Enzyme including BamHI (FD0054, Thermo Fisher Scientific, Rockford, IL, USA), EcoRI (FD0274, Thermo Fisher Scientific, Rockford, IL, USA), HindIII (FD0504, Thermo Fisher Scientific, Rockford, IL, USA), and NdeI (FD0584, Thermo Fisher Scientific, Rockford, IL, USA) and T4 DNA Ligase (FL101-01, Transgen Biotech) was also used to construct donor vectors. The constructed vectors sequence information was shown in Supplementary Data 4.

*Xenopus tropicalis* fertilized eggs were acquired via hormone-induced mating after pre-priming adult male and female frogs with 20 units of human chorionic gonadotropin (HCG) one day in advance and a full dose of 145 units of HCG 2-3 h before collecting the embryos. To induce gene

disruption or integration, we injected a mixture of Cas9 mRNA (300 picograms per egg) and gRNA (200 picograms per egg), or Cas9 mRNA (300 picograms per egg), gRNA (200 picograms per egg), and a donor vector (20 picograms per egg) into the animal pole of fertilized eggs at a volume of 2 nL per egg.

To assess the targeting efficiency, we collected 5–8 healthy embryos at random, 24 h post-microinjection, and proceeded to extract and amplify genomic DNA of the targeted region via PCR. Subsequently, we performed TA cloning by inserting the purified PCR products into the pMD18-T vector (Takara, Katsu, Japan). Sanger DNA sequencing analysis was then conducted on randomly selected single colonies to identify any Indel mutations. Genotyping of adult frogs was performed using small pieces of nails, and the primer sequences used for the PCR genotyping were listed in Supplementary Data 3.

The CRISPOR tool was employed for off-target prediction analysis[40]. To detect off-target effects, we focused on potential sites with a maximum of four mismatches, as detailed in Supplementary Data 1 and Supplementary Data 2. The resulting potential off-target sites were identified and genotyped by Sanger sequencing, and corresponding PCR primers were provided in Supplementary Data 2.

**Hematoxylin-Eosin (H&E) staining, immunofluorescence, whole-mount in situ hybridization (WISH) and transmission electron microscopy (TEM)**

The H&E staining procedure was carried out in accordance with our previously published methodology[60]. Detailedly, *Xenopus tropicalis* tissues were excised and fixed in FAS eyeball fixative (Servicebio, Wuhan, China) at room temperature for 24 h. Following fixation, the samples underwent a series of dehydration steps in ascending concentrations of ethanol (75%, 85%, 95%, and 100%) for 45 min each, and were then subjected to two rounds of xylene treatment, for 1 and 3 h respectively, before being embedded in paraffin wax. The paraffin-embedded samples were sliced into 6 μm thick sections and underwent dewaxing in xylene thrice for 5 min each,

**Article**

and subsequently rehydrated with 100%, 95%, and 70% ethanol for 5 min each, prior to H&E staining with the H&E staining Kit (Baso, Zhuhai, China). For immunofluorescence, the rehydrated sections underwent destaining with a solution containing 0.45% $H_2O_2$ and 1X Tris/Tricine/SDS (T1165, Sigma-Aldrich®) for 15 min at 80 °C to remove melanin pigmentations. The sections were then washed twice with 1X PBS for 5 min each to remove the destaining solution. Antigen retrieval of the destaining sections was performed using citrate antigen retrieval solution (8.1 mM trisodium citrate dihydrate and 1.9 mM citric acid) for 12 h at 60 °C, followed by washing with 1 X PBS solution twice for 5 min each to remove the citrate antigen retrieval solution. The following primary antibodies were then employed: anti-Tyrosinase (ab180753, Abcam), anti-EGFP (ab184601, Abcam), anti-CD31 (ab5690, Abcam), and anti-ERK1 + ERK2 (184699, Abcam). Additionally, two secondary antibodies were utilized: Goat Anti-Mouse IgG H&L (Alexa Fluor® 488) (ab150113, Abcam) and Goat Anti-Rabbit IgG H&L (Alexa Fluor® 594) (ab150080, Abcam). Mounting Medium With DAPI – Aqueous & Fluoroshield (ab104139, Abcam) was employed to stain the cell nucleus. Fluorescence imaging was conducted using a Leica SP8 confocal microscope (TCS SP8, Leica), while H&E-stained sections were visualized with an Olympus bx53 upright microscope (Olympus, Japan).

Regarding WISH, *Xenopus tropicalis* embryos were fixed in MEMFA solution (0.1 M MOPS pH 7.4, 2 mM EDTA, 3.7% Formaldehyde) for 1 h at room temperature and preserved in ethanol at −20 °C. The digoxigenin-labeled antisense probe was synthesized by employing digoxigenin RNA-labeling mix (Roche) and T7 RNA polymerase (M0251S, New England Biolabs) based on a PCR-amplified template containing the T7 promoter. The procedure of WISH was conducted following the established protocol of Harland[70], with certain alterations documented in Hollemann et al.[71]. The relevant PCR primers for the probes were listed in Supplementary Data 3.

As for TEM, a 1 mm × 1 mm section of skin tissue was extracted from frogs and immersed in 2% glutaraldehyde solution for 12 h at 4 °C to fix the sample. Subsequently, the tissue was washed four times with 10 mM PBS for 10 min each time, followed by fixation in 1% osmium tetroxide solution at room temperature for 3 h. The sample was again washed twice with 10 mM PBS for 10 min each time and then rinsed twice with distilled water for 10 min each time. The sample was then treated with 2% uranyl acetate solution for 2 h at room temperature or overnight at 4 °C, and washed with distilled water until the solution was clear. To prepare the tissue for embedding, a 30%, 50%, and 75% acetone solution was prepared with distilled water. The sample was sequentially treated with 30% acetone once, 50% acetone once, 75% acetone once, and 100% acetone twice, each for 10 min at room temperature. In addition, the 100% resin was prepared in advance. The method of preparation of 100% resin was to add Epoxy resin 9.8 g, DDSA 5.6 g, NMH 4.6 g, and DMP-30 0.28 mL in order, and then rotate the mixture at room temperature on a rotary mixer for at least 4 h. The 100% resin was diluted with acetone to prepare 25%, 50%, and 75% resin solutions, and the sample was sequentially treated with 25% resin-acetone once, 50% resin-acetone once, 75% resin-acetone once, and 100% resin once, each for 1.5–2 h at room temperature. Finally, the sample was treated with fresh 100% resin overnight at 4 °C, placed in an embedding box, and embedded at 60 °C for 2 days to complete the embedding. To observe and photograph the sample, 70 nm thin slices were cut using an ultramicrotome (EM UC7, Leica) and placed on copper grids (AGH100, Electron Microscopy China). The sample was imaged using a 120KV biology transmission electron microscope (HT7700 Exalens, Hitachi, Japan).

**Statistics and reproducibility**

Data were analyzed utilizing GraphPad Prism version 9 (GraphPad Prism Software, Inc., CA, USA) and expressed as means ± SD. Unpaired two-tailed t-tests were employed to compare two groups, with statistical significance defined at $P < 0.05$. Reproducibility was maintained through a minimum of three replicates. The accompanying figure legend provides comprehensive information on the data processing methodology.

**Reporting summary**

Further information on research design is available in the Nature Portfolio Reporting Summary linked to this article.

## Data availability

All study data are included in the article and/or supplemental information.

## Code availability

Not applicable.

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

## Acknowledgements

We thank Longjian Niu, Qinnan Yan and Jiarong Wu for their technical supports and Dong Liu Lab for their zebrafish and Pingyuan Zou for frog husbandry. This work was supported by Shenzhen Science and Technology Program (JCYJ20210324120205015), Guangdong Provincial Key Laboratory of Cell Microenvironment and Disease Research (2017B030301018), and Shenzhen Key Laboratory of Cell Microenvironment (ZDSYS20140509142721429). Additionally, the authors would like to dedicate this paper in honor of the forthcoming wedding of Dr. Rensen Ran and Miss Hongjiao Wang.

## Author contributions

R.R. and Y.C. conceived the project. R.R. and L.L. performed the experiments and analyzed the data together with T.X., J.H. and H.H. R.R. and Y.C. wrote the manuscript with input from all the authors. All authors read and approved the final manuscript.

## Competing interests

The authors declare no competing interests.
