## [Peer review file · Communications Biology]

Reviewers' comments:

Reviewer #1 (Remarks to the Author):

This paper details the creation of a transparent and immunodeficient *Xenopus tropicalis* transgenic line by triple knockout of *mitf*^{-/-}/*prkdc*^{-/-}/*il2rg*^{-/-}. Similar lines have been generated in zebrafish, but not in *Xenopus*. Based on previous research connecting the *mitf* gene to melanocyte development, the authors generated a novel *mitf*^{-/-} line. This knockout not only decreased melanocyte generation to generate transparent skin, but also affected xanthophore and granular gland development, a novel phenotype newly characterized in this paper. The authors crossed this transparent *mitf*^{-/-} with a newly generated immunodeficient *prkdc*^{-/-}/*il2rg*^{-/-} line. Subsequent skin transfer from a WT animal to the *mitf*^{-/-}/*prkdc*^{-/-}/*il2rg*^{-/-} line showed reduced immune rejection of the transplanted tissue, demonstrating the utility of such a model.

The paper is thorough, however I recommend several essential experiments in order to support the claims currently made in this manuscript. These experiments will further define both the mechanisms of *mitf*^{-/-} involvement in xanthophore and granular gland development as well as quantitatively define skin graft immune acceptance or rejection in WT or triple knockout animals. Finally, I recommend changes to text and figures within the document to encourage readability and understanding.

Major Comments:

1. While the absence of a reliable, *Xenopus*-reactive Mitf antibody is unavoidable, in situ hybridization of *mitf* during larval development should be feasible. This experiment is crucial to definitively show *mitf* expression in xanthophores and granular glands. In fact, the authors mention they analyzed the expression of *mitf* using whole mount in situ hybridization in the text but it is not in the figure (Line 233-234).
2. RT-PCR results documenting the decrease in xanthophore and granular gland-related gene expression is not sufficient, authors should also conduct in situ hybridization at larval stages to observe changes in cell distribution. This will clarify if reductions in expression are uniform, or if some areas have normal expression while others have no expression. If in situ hybridization is not available due to staging (for example, if these cell types emerge later than early larval stages), antibodies would also appropriately visualize whole mount changes in cell composition. These experiments could be complemented by use of the EGFP-BRAFV600E line.
3. Additionally describe, preferably quantitatively, the postulated immune rejection based on T cell number in the *mitf*^{-/-} recipient animals vs *mitf*^{-/-}/*prkdc*^{-/-}/*il2rg*^{-/-} animals. Potentially, utilize arrows or additional visualization via immunohistochemistry staining to definitively show immune cell infiltration and or other indicators of immune rejection.
4. The last experiment investigating dysplastic nevus and metastatic melanoma transplantation only has one animal per group. While compelling, replicates are needed to establish that this observation is reproducible .

Minor Comments:

Figures. There are some labeling errors or confusing organization in figures, detailed below by figure number.

- a. Figure 2B: change order of images: 1.5 months WT, *mitf*^{-/-}, 2 years WT, *mitf*^{-/-}
- b. Figure 3C-F: reorganize to better align WT and *mitf*^{-/-}, current images are incomparable
- c. Figure 4A: WT & *mitf*^{-/-} switched
- d. Figure 5: All fluorescent images should have WT matches
- a. Figure 5G: Should have a WT control
- b. Figure 5F: WT image should be taken at the same resolution and lighting
- e. Figure 9B: Include WT control image

Introduction

1. Overall the authors should reiterate why a transparent and immunodeficient line needs to be created in *Xenopus*, especially since one already exists in zebrafish. What attributes of *Xenopus* make it an ideal animal for such a model? Why is immunodeficiency useful for tumor models? This topic can be specifically addressed in the first two sentences of the abstract (lines 38-49) as well as in the introduction.
2. Within the introduction, introduce what skin cell types are conserved across vertebrates. What

cells are present in *Xenopus*, zebrafish, and mammals?

3. Confirm consistent font (see "Introduction" title on line 81)

4. Line 85- avoid the wording "higher species", tetrapods could instead be substituted.

5. Is there a reason the authors chose not to knock out mpv17 in *Xenopus*?

6. Line 130- not a complete sentence

Results

7. Line 244- The statement that mitf knockout affected xanthophore and granular gland development was unsubstantiated. The RT-PCR analysis confirmed loss of these cell types, but did not elucidate the developmental origin of these defects. This statement would necessitate additional experiments in larvae in order to pinpoint the stage at which mitf KO affects xanthophore and granular gland development- does it affect their original differentiation?

Maintenance?

8. Line 250- what test was conducted to analyze this significance level?

9. Move paragraph (Lines 295-306) to line 261.

10. In lines 330-337- Using the figure, point out specific and potentially quantitative descriptors of skin graft rejection or acceptance.

Reviewer #2 (Remarks to the Author):

The authors generated a colorless, immunodeficient frog line (*Xenopus tropicalis*) using the CRISPR/Cas9 system to disrupt the mitf gene, which regulates melanophores, xanthophores, and granular glands, as well as the prkdc and il2rg genes, which are responsible for the development of T and B cells. This frog line is a promising platform for research in tumor and developmental biology. To demonstrate this concept, the authors performed allogeneic skin transplantation, xanthophoromas, and melanomas, and observed tumor metastasis and migration. Additionally, this study offers a new understanding of the functions of the formation of melanophores, xanthophores, and granular glands.

The manuscript reports in detail the process and methods of generating a new animal model line and successfully demonstrates immunodeficiency by allotransplantation of skin and tumor cells. They also provided new information on the generation of a new animal model, as well as the formation of melanophores, xanthophores, and granular glands under mitf gene function. This reviewer finds the results of this research very interesting and valuable and would like to make a few comments, including the suggestions below.

Major comments:

1) As the authors claimed, I agree that this frog line provides a new platform for vertebrate research. This needs to be brought to the attention not only of amphibian researchers but also of scientists who use existing animal models, such as mice. Therefore, for further development, this reviewer has the impression that some modifications will be necessary to make this manuscript understandable to researchers in many fields. In addition, the text is very intricate. From this perspective, this reviewer considers it preferable to devise a way to help readers intuitively understand the content. For example, what about a table summarizing the gene deletion types (frogs with mitf deletion and frogs with three gene deletions: mitf, prkdc, and il2rg) and phenotypes, as well as depicting the relationship between each deleted gene and gene function as a visual abstract?

2) Although immunodeficiency of 3-gene-deficient frogs has been described in detail in allogeneic xenograft experiments, if I am not misreading, no analysis has been presented on the reduction in the number of immune cells, such as B cells, T cells, and dendritic cells, or on the suppression of their function. Since the development of T and B lymphocytes was impaired in zebrafish in which the prkdc gene was altered, we can imagine that *Xenopus tropicalis* in this manuscript is similarly immunosuppressed, but there is no direct evidence of an actual lymphocyte defect. (Underdeveloped thymus?) Please provide relevant data such as peripheral blood cell analysis or tissue expression of genes specific to immunocompetent cells. The failure of complete xenotransplantation between *Xenopus* and zebrafish in this study, if not graft failure due to the optimal osmotic pressure of the graft, could be deeply related to these factors.

3) Similarly, the IL2RG gene is a shared receptor subunit for six cytokines (IL-4, IL-7, IL-9, IL-15, and IL-21); however, orthologous genes in *Xenopus tropicalis* have not been described. Please describe the presence or absence of orthologous genes for the ligands of these shared receptors and their expression in frogs lacking these three genes.

4) Wild glasses with spectacular transparency have recently been reported, and the authors cited them in Reference 3 (Taboada, C. et al. *Science*. 2022;378(6626):1315-1320.) The frogs in Reference 3 are not *Xenopus tropicalis* and should be corrected accordingly. In the case of this glass frog, red blood cells in the blood circulation inhibit transparency. Therefore, in the case of *mitf*^{-/-} frogs, it would be interesting to know whether such inhibition of transparency by red blood cells was observed.

5) I may have missed it, but I couldn't find a description of the rearing of immunodeficient frogs. Quality control of rearing water is vital for immunocompromised aquatic animals. This information is essential for those who intend to experiment with these frogs.

Minor comments.

1) The authors stated that the secretion of antimicrobial substances might be degraded due to failure of the formation of granular glands in the *mitf*^{-/-} frog. As the authors note, the *mitf* gene is involved in the formation of immunocompetent cells in humans (Ref. 55). Therefore, it is possible that *mitf*^{-/-} frogs functionally suppress the immune system and attenuate immunosuppressive effects. The authors need to discuss this possibility.

2) In Figure 3H, Mel is almost entirely missing from the field of view. It is better to expand the photographic area. Similarly, MG and *S. bas* in Fig. 3F and BC are shown in Fig. 3H, respectively.

Reviewer #3 (Remarks to the Author):

Ran et al., have developed a functionally nullifying biallelic *mitf* mutations that successfully created a colorless line of *Xenopus tropicalis*. This novel line displays transparent skin and visible internal organs within two months post-metamorphosis and lacks cutaneous melanophores throughout its lifespan. This newly developed amphibian model holds great potential for use in tumorous and developmental studies.

Questions and suggestions concerning some details of the manuscript.

1. Include information from the literature about Mitf signaling pathways. For example, it is known that Wnt signaling (which must be included) which is very important in many biological processes plays a critical role in melanocyte regulation.

1. According to the literature GSK3, downstream from both the PI3K and Wnt pathways, and BRAF/MAPK signaling converges to control MITF. Did the authors test the interaction between Wnt and Wnt/GSK3? These possible interactions and regulation systems should be explain in the discussion.

A culture cell experiment should be included showing a gain-of-function situation of the the *mitf*^{-/-} /*prkdc*^{-/-} /*il2rg*^{-/-}

Are β -catenin or Wnt target genes affected in the RT-PCR analysis?

Other signals control MITF activity through posttranslational modification. The authors should mention this in the discussion as well.

I would recommend to include in the citations the following papers:

It will be helpful to include a possible signaling pathway for MITF regulation.

<https://doi.org/10.1073/pnas.1810498115>

K Takeda, et al., Ser298 of MITF, a mutation site in Waardenburg syndrome type 2, is a phosphorylation site with functional significance. *Hum Mol Genet* 9, 125–132 (2000).

<https://www.pnas.org/doi/10.1073/pnas.1810498115>

D Ploper, et al., MITF drives endolysosomal biogenesis and potentiates Wnt signaling in melanoma cells. *Proc Natl Acad Sci USA* 112, E420–E429 (2015).

RI Dorsky, DW Raible, RT Moon, Direct regulation of nacre, a zebrafish MITF homolog required for pigment cell formation, by the Wnt pathway. *Genes Dev* 14, 158–162 (2000).

Albrecht, L. V., Tejada-Muñoz, N., & De Robertis, E. M. (2021). Cell Biology of Canonical Wnt Signaling. *Annual review of cell and developmental biology*, 37, 369–389.

<https://doi.org/10.1146/annurev-cellbio-120319-023657>

Responses to the Reviewers

We thank the reviewers for their comments and suggestions. We have taken these comments into account in the revision of the manuscript.

Reviewer #1

This paper details the creation of a transparent and immunodeficient *Xenopus tropicalis* transgenic line by triple knockout of *mitf*^{-/-}/*prkdc*^{-/-}/*il2rg*^{-/-}. Similar lines have been generated in zebrafish, but not in *Xenopus*. Based on previous research connecting the *mitf* gene to melanocyte development, the authors generated a novel *mitf*^{-/-} line. This knockout not only decreased melanocyte generation to generate transparent skin, but also affected xanthophore and granular gland development, a novel phenotype newly characterized in this paper. The authors crossed this transparent *mitf*^{-/-} with a newly generated immunodeficient *prkdc*^{-/-}/*il2rg*^{-/-} line. Subsequent skin transfer from a WT animal to the *mitf*^{-/-}/*prkdc*^{-/-}/*il2rg*^{-/-} line showed reduced immune rejection of the transplanted tissue, demonstrating the utility of such a model.

The paper is thorough, however I recommend several essential experiments in order to support the claims currently made in this manuscript. These experiments will further define both the mechanisms of *mitf*^{-/-} involvement in xanthophore and granular gland development as well as quantitatively define skin graft immune acceptance or rejection in WT or triple knockout animals. Finally, I recommend changes to text and figures within the document to encourage readability and understanding.

Major Comments:

1. While the absence of a reliable, *Xenopus*-reactive Mitf antibody is unavoidable, in situ hybridization of *mitf* during larval development should be feasible. This experiment is crucial to definitively show *mitf* expression in xanthophores and granular glands. In fact, the authors mention they analyzed the expression of *mitf* using whole mount in situ hybridization in the text but it is not in the figure (Line 233-234).

➤ *We have incorporated the in situ hybridization results for mitf into Fig. 4A, Supplementary Fig. 13, and Supplementary Fig. 14. The depicted in situ hybridization findings for mitf predominantly illuminated the developmental trajectory of melanophores. Regrettably, the investigation into the developmental course of Xenopus tropicalis xanthophores remains limited. Drawing upon conjecture derived from the developmental course of Xenopus laevis xanthophores, it is posited that the emergence of Xenopus tropicalis xanthophores may commence around stage 46 (<https://www.xenbase.org/xenbase/>). Therefore, we attempted in situ hybridization experiments on Xenopus tropicalis embryos beyond stage 40. Unfortunately, despite*

multiple attempts, we consistently failed to detect any hybridization signals for *mitf* mRNA (see **Responses Figure.1**).

Responses Figure.1 Results of *mitf* in situ hybridization in WT *Xenopus tropicalis* embryos were presented. Subsection **A** denoted positive *mitf* in situ hybridization results, while subsection **B** revealed the absence of positive signals of *mitf* in situ hybridization during stages 40-48 of *Xenopus tropicalis*.

Gland rudiments in *Xenopus laevis* manifest from stage 57 onwards, culminating in well-developed granular glands by stage 63¹⁻³. We postulated a comparable developmental trajectory for granular glands in *Xenopus tropicalis*. However, despite multiple endeavors, *mitf* mRNA in situ hybridization signals were not discernible in *Xenopus tropicalis* skin specimens. To elucidate the expression of *mitf* in xanthophores and granular glands, we drew upon findings from a parallel research initiative involving the targeted insertion of EGFP-BRAF^{V600E} into the *mitf* locus of *Xenopus tropicalis*, denoted as *mitf*-BRAF^{V600E} (depicted in **Figure. 6A**). In accordance with the targeted insertion design of EGFP-BRAF^{V600E} into the *mitf* locus, transcripts containing the last exon of *mitf* are co-expressed with EGFP-BRAF^{V600E} transcripts, thereby initiating the MAPK signaling pathway and resulting in corresponding phenotypic changes. As expected, in *mitf*-BRAF^{V600E} frogs, the expression of EGFP-BRAF^{V600E} activated the MAPK signaling pathway, as depicted in **Figures.6D-E** of the manuscript.

This observation suggested that the significant increase in xanthophores in mitf-BRAF^{V600E} frogs was attributable to mitf expression in these cells, as illustrated in Figures.6B, 6F, and 6G. Furthermore, the diminished number of granular glands in the skin of mitf-BRAF^{V600E} frogs implied that the targeted insertion of EGFP-BRAF^{V600E} into the mitf locus resulted in abnormal granular gland development. Crucially, in mitf^{-/-} frogs, the absence of melanocytes, xanthophores, and granular glands was evident, as illustrated in Figure.3 of the manuscript. Conversely, rescue frogs, with targeted restoration at the mitf knockout site, demonstrated the re-establishment of melanocytes and xanthophores in G0 mosaic skin, as depicted in Figures.5B-D of the manuscript. Concurrently, a notable presence of gland rudiments, potentially serving as granular gland precursors, was observed in the G0 mosaic skin of rescue frogs, as shown in Figure.5D. This collective evidence strongly supported the contention that mitf played a pivotal role in the development of melanocytes, xanthophores, and granular glands. Regrettably, the current absence of direct evidence for mitf expression in xanthophores and granular glands in Xenopus tropicalis can be attributed to challenges in implementing in situ hybridization techniques in tadpoles and tissue samples, coupled with the unavailability of effective Mitf antibodies. Nevertheless, future investigations leveraging single-molecule fluorescence in situ hybridization (smFISH) techniques and employing highly effective antibodies are poised to furnish the most direct confirmation of mitf expression in xanthophores and granular glands. (The response and modifications to this issue can be found in the manuscript on Fig. 4A, Supplementary Fig. 11, and Supplementary Fig. 12 for detailed information.)

2. RT-PCR results documenting the decrease in xanthophore and granular gland-related gene expression is not sufficient, authors should also conduct in situ hybridization at larval stages to observe changes in cell distribution. This will clarify if reductions in expression are uniform, or if some areas have normal expression while others have no expression. If in situ hybridization is not available due to staging (for example, if these cell types emerge later than early larval stages), antibodies would also appropriately visualize whole mount changes in cell composition. These experiments could be complemented by use of the EGFP-BRAF^{V600E} line.

➤ *Employing the expression patterns of xanthophore and granular gland marker genes to signify the comprehensive absence of xanthophores and granular glands in mitf^{-/-} Xenopus tropicalis, though informative, remains somewhat insufficient. As stated in response to point "I" the application of in situ hybridization techniques in tadpoles proves challenging, and effective antibodies are lacking. Consequently, we chose to address this challenge by examining marker gene expression. Indeed, attempts were made to investigate these inquiries using mitf-BRAF^{V600E} Xenopus tropicalis (referred*

to as the EGFP-BRAF^{V600E} line). In *mitf*-BRAF^{V600E} frogs, we performed immunofluorescence co-staining using multiple antibodies targeting xanthophores in conjunction with the EGFP antibody. The outcomes revealed the unreliability of commercially available xanthophore-targeting antibodies in *Xenopus tropicalis*. In other words, while the EGFP antibody demonstrated efficacy, the absence of correspondingly reliable antibodies for xanthophore labeling hindered co-localization studies. Comparable challenges were encountered in immunofluorescence staining for granular glands. To comprehensively assess the absence of granular glands and xanthophores in the skin of *mitf*^{-/-} *Xenopus tropicalis*, we employed characteristics delineated in the literature pertaining to granular gland^{1,4} and xanthophores^{5,6} in *Xenopus laevis*. Assessment of the entire-body granular glands and xanthophores was conducted in both WT and *mitf*^{-/-} *Xenopus tropicalis*. The findings, illustrated in **Responses Figure.2**, demonstrated the lack of spontaneously fluorescing green xanthophores in the tadpoles of *mitf*^{-/-} *Xenopus tropicalis*. Additionally, the granular glands were entirely absent in the skin of *mitf*^{-/-} frogs. Consequently, the aforementioned data unequivocally signify the comprehensive absence of xanthophores and granular glands in the skin of *mitf*^{-/-} *Xenopus tropicalis*. (The response and modifications to this issue can be found in the manuscript on Line 233-239 and Supplementary Fig.9 for detailed information.)

Responses Figure.2 Xanthophores and granular glands were notably absent in the skin of *mitf*^{-/-} *Xenopus tropicalis*. **A**, The intrinsic green fluorescence indicative of xanthophores was not observed in stage 49 and stage 57 *mitf*^{-/-} *Xenopus tropicalis*. **B**, The progression of granular gland development from stage 57 to stage 59 in both WT and *mitf*^{-/-} *Xenopus tropicalis*, with white arrows denoting the emergence of developing granular glands. **C**, the development of fully matured granular glands on the skin of WT and *mitf*^{-/-} *Xenopus tropicalis* nearing completion of metamorphosis, with white arrows indicating well developed granular glands. **A**, the scale bar is 0.5 mm. **B**, the scale bar is 50 μm. For the photographs of frogs in **C**, the scale bar is 1 mm, while the remaining images have a scale bar of 50 μm.

3. Additionally describe, preferably quantitatively, the postulated immune rejection based on T cell number in the *mitf*^{-/-} recipient animals vs *mitf*^{-/-}/*prkdc*^{-/-}/*il2rg*^{-/-} animals. Potentially, utilize arrows or additional visualization via immunohistochemistry staining to definitively show immune cell infiltration and or other indicators of immune rejection.

➤ In the skin allograft experiments, we employed immunofluorescence experiments utilizing the T cell marker CD3 to delineate T cell infiltration. As illustrated in **Responses-Figure.3**, a notable infiltration of CD3⁺ cells was observed in the skin of *mitf*^{-/-} recipients, contrasting with the absence of such infiltration in the skin of *mitf*^{-/-}/*prkdc*^{-/-}/*il2rg*^{-/-} recipients. This observation suggested an immune deficiency in *mitf*^{-/-}/*prkdc*^{-/-}/*il2rg*^{-/-} frogs, resulting in a significantly attenuated immune rejection response to skin grafts. Furthermore, in the corresponding H&E staining and CD3 immunofluorescence images, white arrows were employed to denote the presence of infiltrating T cells. (The response and modifications to this issue can be found in the manuscript on Line 381, Fig.8 for detailed information.)

Responses Figure.3 T cell infiltration following skin allografts in *mitf*^{-/-} and *mitf*^{-/-}/*prkdc*^{-/-}/*il2rg*^{-/-} *Xenopus tropicalis*. T cells were labeled with CD3, and cell nuclei were stained with DAPI. The recipient *mitf*^{-/-} and *mitf*^{-/-}/*prkdc*^{-/-}/*il2rg*^{-/-} *Xenopus tropicalis* were both 6 months old, and the donor skin was obtained from the dorsal skin of 1-year-old WT *Xenopus tropicalis*. The scale bar is 50 μm.

4. The last experiment investigating dysplastic nevus and metastatic melanoma transplantation only has one animal per group. While compelling, replicates are needed to establish that this observation is reproducible.

➤ *Regrettably, due to constraints in the availability of tumor samples, we were constrained to conduct an additional set of allograft experiments involving dysplastic nevi on $mitf^{-/-}$ and $mitf^{-/-}/prkdc^{-/-}/il2rg^{-/-}$ *Xenopus tropicalis*. The donor dysplastic nevi were obtained from $cdkn2b^{-/-}/tp53^{-/-}$ *Xenopus tropicalis*. As depicted in **Responses Figure.4**, the dysplastic nevus transplanted onto the skin of $mitf^{-/-}$ *Xenopus tropicalis* exhibited evident signs of rejection on the 40th day post-transplantation. In contrast, the dysplastic nevus transplanted onto the skin of $mitf^{-/-}/prkdc^{-/-}/il2rg^{-/-}$ *Xenopus tropicalis* from the same batch did not manifest any signs of rejection at this juncture. However, no evidence of melanophore migration was discerned on the skin of both $mitf^{-/-}$ and $mitf^{-/-}/prkdc^{-/-}/il2rg^{-/-}$ *Xenopus tropicalis*. This may be attributable to the advanced age of the recipient frogs (1-year-old adults). Further exploration and discourse on this matter can be pursued in the future with an adequate number of tumor samples. In summary, this transplantation experiment not only reaffirmed the applicability of $mitf^{-/-}/prkdc^{-/-}/il2rg^{-/-}$ *Xenopus tropicalis* for tumor transplantation but also suggests a comparatively limited metastatic potential of dysplastic nevi from $cdkn2b^{-/-}/tp53^{-/-}$ *Xenopus tropicalis*. (The response and modifications to this issue can be found in the manuscript on Line 430-436, Supplementary Fig.30 for detailed information.)*

Responses Figure.4 *The findings from the transplantation of dysplastic nevi in $mitf^{-/-}$ and $mitf^{-/-}/prkdc^{-/-}/il2rg^{-/-}$ *Xenopus tropicalis* were presented. Donor dysplastic nevi were obtained from 14-month-old $cdkn2b^{-/-}/tp53^{-/-}$ *Xenopus tropicalis*, which spontaneously developed nevi. Two distinct dysplastic nevi samples were individually grafted onto the dorsal skin of one-year-old $mitf^{-/-}$ and $mitf^{-/-}/prkdc^{-/-}/il2rg^{-/-}$ *Xenopus tropicalis*. The accompanying images depict outcomes observed on the 40th day post-transplantation, with black arrows highlighting the transplanted dysplastic nevi. The scale is 1 mm.*

Minor Comments:

Figures. There are some labeling errors or confusing organization in figures, detailed below by figure number.

- a. Figure 2B: change order of images: 1.5 months WT, *mitf*^{-/-}, 2 years WT, *mitf*^{-/-}
- b. Figure 3C-F: reorganize to better align WT and *mitf*^{-/-}, current images are incomparable
- c. Figure 4A: WT & *mitf*^{-/-} switched
- d. Figure 5: All fluorescent images should have WT matches
- a. Figure 5G: Should have a WT control
- b. Figure 5F: WT image should be taken at the same resolution and lighting
- e. Figure 9B: Include WT control image

➤ *Thank you for pointing out these issues. They have been addressed and corrected in the manuscript. (Please refer to Fig. 2, Fig. 3, Fig. 4, Fig.6 and Fig.9 for details.)*

Introduction

1. Overall the authors should reiterate why a transparent and immunodeficient line needs to be created in *Xenopus*, especially since one already exists in zebrafish. What attributes of *Xenopus* make it an ideal animal for such a model? Why is immunodeficiency useful for tumor models? This topic can be specifically addressed in the first two sentences of the abstract (lines 38-49) as well as in the introduction.

➤ *This issue has been addressed and corrected in the Abstract of the manuscript. (Please refer to Line 38 – Line 44, Line 82-99 for details.)*

2. Within the introduction, introduce what skin cell types are conserved across vertebrates. What cells are present in *Xenopus*, zebrafish, and mammals?

➤ *This issue has been addressed and corrected in the Introduction of the manuscript. (Please refer to Line 99 – Line 104 for details.)*

3. Confirm consistent font (see “Introduction” title on line 81)

➤ *This has been corrected in the manuscript. (Please refer to Line 81 for details.)*

4. Line 85- avoid the wording “higher species”, tetrapods could instead be substituted.

➤ *The manuscript has avoided such a description. (Please refer to Line 82 – Line 99 for details.)*

5. Is there a reason the authors chose not to knock out *mpv17* in *Xenopus*?

➤ *In 1990, Hans Weiher and colleagues reported that adult mice with homozygous mutations in the *Mpv17* gene developed nephrotic syndrome and chronic renal failure, resulting in a survival rate of less than 10% at six months⁷. In humans, the *MPV17* gene encodes a mitochondrial inner membrane protein (*MPV17*), and its dysfunction or loss*

can cause rare autosomal recessive disorders, specifically mitochondrial DNA depletion syndromes^{8,9}. While in zebrafish, the orthologous gene of human MPV17 is transparent (*tra*), also known as *mpv17*. Mutations in this gene can lead to the complete loss or significant reduction of iridophores in zebrafish^{10,11}. However, research suggested that the precise contribution of *mpv17* to the transparent skin phenotype in casper zebrafish warrants further investigation¹². Considering these factors, and to ensure optimal survival rates and minimal impact on health in the final construction of transparent immunodeficient frogs, we have chosen to initiate model development by attempting to knock out the *mitf* gene in *Xenopus tropicalis*. This decision was further explained in the Introduction section of the manuscript. (*The response and modifications to this issue can be found in the manuscript on Line 109 – Line 117 for details.*)

6. Line 130- not a complete sentence

➤ *This has been corrected in the manuscript. (Please refer to Line 146 – Line 148 for details.)*

Results

7. Line 244- The statement that *mitf* knockout affected xanthophore and granular gland development was unsubstantiated. The RT-PCR analysis confirmed loss of these cell types, but did not elucidate the developmental origin of these defects. This statement would necessitate additional experiments in larvae in order to pinpoint the stage at which *mitf* KO affects xanthophore and granular gland development- does it affect their original differentiation? Maintenance?

➤ *The conservation of Mitf's role in melanocyte development among vertebrates has been substantiated in this manuscript. Regrettably, the confirmation of anomalies in *mitf*^{-/-} frog xanthophores and granular gland development through in situ hybridization is currently unattainable. Consequently, we examined the development of *mitf*^{-/-} frog xanthophores by leveraging the inherent green fluorescence of these cells^{5,6}. Illustrated in **Responses Figure.2A**, starting from stage 49, xanthophores with spontaneously branching dendritic morphologies, emitting green fluorescence, emerged on the dorsal side of WT tadpole heads, exhibiting a dispersed clustering pattern. By stage 57, xanthophores in WT tadpoles were ubiquitously distributed across the body. Nevertheless, xanthophores were conspicuously absent in *mitf*^{-/-} frogs throughout the entire developmental sequence (**Responses Figure.2A**). Therefore, integrating the data presented in the manuscript, what can be conclusively determined is the absence of mature xanthophore development in *mitf*^{-/-} *Xenopus tropicalis*. Unfortunately, as for the development of *mitf*^{-/-} frog xanthophores, we currently lack experimental evidence to provide a definitive answer. Notably, research on xanthophore development in *Xenopus**

tropicalis remains relatively limited. Consequently, insights into the developmental trajectory of *Xenopus tropicalis* xanthophores were sought through a comparative analysis with the well-documented developmental process of zebrafish xanthophores¹³. Investigating the zebrafish xanthophore developmental process reveals that neural crest cells expressing key genes such as *pax3*, *sox9*, *ltk*, *sox10*, *pax7*, *tfec*, and *mitfa* differentiate into MIX progenitor cells¹³. These precursors possess the capacity to further develop into melanocytes, xanthophores, and iridophores. Subsequently, MIX progenitor cells modulate gene expression, downregulating *mitfa* while upregulating *sox10* and *pax7*, ultimately giving rise to xanthoblasts, which further differentiate into xanthophores¹³. The developmental trajectory of xanthophores in *Xenopus tropicalis* likely shares a degree of conservation with that of zebrafish xanthophores. Integrating the manuscript data, a hypothesis was formulated that in *mitf*^{-/-} frogs, neural crest cells underwent differentiation into MX progenitor cells, which had the potential to progress into melanocytes and xanthophores. However, the functional loss of *mitf* impeded the sustainability and viability of MX progenitor cells, inferred from *mitf*'s known role in melanocyte development¹⁴, resulting in the observed absence of mature melanocytes and xanthophores in *mitf*^{-/-} frogs. The investigation into *mitf*'s role in xanthophore development in *Xenopus tropicalis*, alongside the broader context of pigment cell development in these frogs, represents a meaningful research question. In forthcoming studies, we anticipate that either our team or other researchers will contribute valuable insights to achieve a closer approximation of the underlying truth.

Based on the characteristics reported in the literature regarding the granular gland^{1,4}, we conducted an examination of granular gland development in both WT and *mitf*^{-/-} *Xenopus tropicalis*. As illustrated in **Responses Figure.2B-C**, at stage 59, granular glands were discernible on the skin of WT frogs, whereas no granular glands were evident on the skin of *mitf*^{-/-} frogs. This observation indicated that the depletion of *mitf* results in the comprehensive absence of granular glands throughout the entire body of *Xenopus tropicalis*. Furthermore, we ascertained that in WT *Xenopus tropicalis* at stage 58, certain cells in the basal layer of the skin epidermis underwent development, giving rise to gland rudiments (**Responses Figure.5A**). These gland rudiments subsequently underwent maturation, evolving into granular glands and mucous glands by stage 59, with further developmental progress and maturation evident at stage 60 (**Responses Figure.5A**). In contrast, gland rudiments in *mitf*^{-/-} frogs underwent development, but the emergence of granular glands did not occur (**Responses Figure.5B**). This suggested that granular glands in *mitf*^{-/-} *Xenopus tropicalis* might only advance to a stage resembling "progenitor cells." Consequently, the absence of *mitf* function resulted in a subsequent deficiency in mature granular glands. Unfortunately, due to the challenging implementation of mRNA in situ hybridization techniques at this developmental stage in *Xenopus tropicalis* and the unavailability of antibodies, we are

currently unable to furnish more precise experimental data to elucidate the spatiotemporal information on the developmental defects of granular glands in *mitf*^{-/-} *Xenopus tropicalis*. The deletion of *Mitf* led to the absence of mature granular glands, representing a novel phenotype reported for the first time. The existing literature on the development of granular glands in *Xenopus tropicalis* is limited. Therefore, concerning the role of *Mitf* in granular gland development, as future investigations into the development of granular glands in *Xenopus tropicalis* progress, more comprehensive answers are anticipated.

(The response and modifications to this issue can be found in the manuscript on Line 233-239, Supplementary Fig.8 for detailed information.)

Responses Figure.5 Development of glands in WT and *mitf*^{-/-} *Xenopus tropicalis*. A, The development of glands in WT frogs from stage 58 to stage 60 was illustrated. B, The development of glands in *mitf*^{-/-} frogs from stage 58 to stage 60 was delineated. GG, granular gland; GR, gland rudiment; MG, mucous gland. The scale is 60 μ m.

7. Line 250- what test was conducted to analyze this significance level?

➤ *We apologize for the inaccurate description. The error in the corresponding section of the manuscript has been rectified. Relative gene expression levels were determined based on the intensity of bands in agarose gel electrophoresis. Grayscale values were not analyzed for quantitative comparison. Presenting and interpreting RT-PCR results in this manner has been accepted in our previous studies¹⁵⁻¹⁷ and is also demonstrated in published research by others^{18,19}. Therefore, we have chosen to continue with this presentation format. Thank you for your support and understanding. (Please refer to Line 266, Line 275, Line 282 for details.)*

9. Move paragraph (Lines 295-306) to line 261.

➤ *This has been corrected in the manuscript (Please refer to Line 285 - Line 297 for details.)*

10. In lines 330-337- Using the figure, point out specific and potentially quantitative descriptors of skin graft rejection or acceptance.

➤ *This issue has been addressed in the manuscript. (Please refer to Line 361 – Line 366 for details.)*

Reviewer #2

The authors generated a colorless, immunodeficient frog line (*Xenopus tropicalis*) using the CRISPR/Cas9 system to disrupt the *mitf* gene, which regulates melanophores, xanthophores, and granular glands, as well as the *prkdc* and *il2rg* genes, which are responsible for the development of T and B cells. This frog line is a promising platform for research in tumor and developmental biology. To demonstrate this concept, the authors performed allogeneic skin transplantation, xanthophoromas, and melanomas, and observed tumor metastasis and migration. Additionally, this study offers a new understanding of the functions of the formation of melanophores, xanthophores, and granular glands.

The manuscript reports in detail the process and methods of generating a new animal model line and successfully demonstrates immunodeficiency by allotransplantation of skin and tumor cells. They also provided new information on the generation of a new animal model, as well as the formation of melanophores, xanthophores, and granular glands under *mitf* gene function. This reviewer finds the results of this research very interesting and valuable and would like to make a few comments, including the suggestions below.

Major comments:

1) As the authors claimed, I agree that this frog line provides a new platform for

vertebrate research. This needs to be brought to the attention not only of amphibian researchers but also of scientists who use existing animal models, such as mice. Therefore, for further development, this reviewer has the impression that some modifications will be necessary to make this manuscript understandable to researchers in many fields. In addition, the text is very intricate. From this perspective, this reviewer considers it preferable to devise a way to help readers intuitively understand the content. For example, what about a table summarizing the gene deletion types (frogs with *mitf* deletion and frogs with three gene deletions: *mitf*, *prkdc*, and *il2rg*) and phenotypes, as well as depicting the relationship between each deleted gene and gene function as a visual abstract?

➤ We fully agree with this point. To highlight the main content of our manuscript more effectively, we have presented it in a graphical abstract, as shown in **Responses Figure.6**. (The response and modifications to this issue can be found in the manuscript on Line 444, Fig.11 for detailed information.)

Responses Figure.6 Schematic of *mitf*^{-/-}/*prkdc*^{-/-}/*il2rg*^{-/-} *Xenopus tropicalis* construction. The process revealed the crucial role of *mitf* in the development of melanophores, xanthophores, and granular glands. The resulting *mitf*^{-/-}/*prkdc*^{-/-}/*il2rg*^{-/-} *Xenopus tropicalis* was suitable for research on allograft tumor transplantation.

2) Although immunodeficiency of 3-gene-deficient frogs has been described in detail in allogeneic xenograft experiments, if I am not misreading, no analysis has been

presented on the reduction in the number of immune cells, such as B cells, T cells, and dendritic cells, or on the suppression of their function. Since the development of T and B lymphocytes was impaired in zebrafish in which the *prkdc* gene was altered, we can imagine that *Xenopus tropicalis* in this manuscript is similarly immunosuppressed, but there is no direct evidence of an actual lymphocyte defect. (Underdeveloped thymus?) Please provide relevant data such as peripheral blood cell analysis or tissue expression of genes specific to immunocompetent cells. The failure of complete xenotransplantation between *Xenopus* and zebrafish in this study, if not graft failure due to the optimal osmotic pressure of the graft, could be deeply related to these factors.

➤ *To enhance this data, we utilized qPCR to evaluate T-cell, B-cell, and NK-cell marker gene expression in spleen, lung, liver, and blood specimens from both WT and $prkdc^{-}/il2rg^{-}$ adult frogs. Given the unavailability of suitable antibodies, qPCR was employed for characterizing the status of T-cells, B-cells, and NK-cells. As shown in **Responses Figure.7**, the expression of T-cell markers *cd3g* and *cd8b* significantly decreased to nearly negligible levels, and B-cell markers *cd19* and *ms4a1* exhibited notable reductions (**Responses Figure.7A and B**). In contrast, the expression of NK-cell markers *cxcr4* and *cd59* displayed a more intricate pattern (**Responses Figure.7C**). In essence, the data indicate that, despite significant impacts on T-cells, B-cells, and NK-cells in $prkdc^{-}/il2rg^{-}$ tropical clawed frogs, there is still residual expression of T-cell, B-cell, and NK-cell marker genes. This suggests that $prkdc^{-}/il2rg^{-}$ and $mitf^{-}/prkdc^{-}/il2rg^{-}$ frogs may maintain certain immune capabilities mediated by T-cells, B-cells, and NK-cells despite immune system defects. This aligns with the findings of zebrafish skin transplants in the manuscript, where zebrafish skin grafts in $prkdc^{-}/il2rg^{-}$ frogs exhibited temporary survival before rejection. Unfortunately, current experimental data are insufficient to elucidate the influence of osmotic pressure on skin transplant outcomes. Nonetheless, considering the typical extracellular osmotic pressure of vertebrate cells falls within the range of 270–300 mOsm²⁰, we believe that immune rejection is the primary factor contributing to the unsuccessful zebrafish skin transplants. (The response and modifications to this issue can be found in the manuscript on Line 353-356, Supplementary Fig. 23, Supplementary Fig. 24 for detailed information.)*

Responses Figure.7 The expression of T-cell (A), B-cell (B), and NK-cell (C) marker genes in *prkdc*^{-/-}/*il2rg*^{-/-} (PI) *Xenopus tropicalis* is shown.

3) Similarly, the *IL2RG* gene is a shared receptor subunit for six cytokines (IL-4, IL-7, IL-9, IL-15, and IL-21); however, orthologous genes in *Xenopus tropicalis* have not been described. Please describe the presence or absence of orthologous genes for the ligands of these shared receptors and their expression in frogs lacking these three genes.

➤ **In searching the NCBI database, we found annotations for five out of the six cytokines (IL-2, IL-4, IL-7, IL-9, IL-15, and IL-21) in *Xenopus tropicalis*, namely *il21*, *il9*, *il15*, *il2*, and *il4*. Using qPCR and RT-PCR, we compared the expression of these five genes in the spleen of WT and *prkdc*^{-/-}/*il2rg*^{-/-} *Xenopus tropicalis*. As depicted in **Responses Figure.8**, *il2* exhibited a significant decrease in expression in the spleen of *prkdc*^{-/-}/*il2rg*^{-/-} *Xenopus tropicalis*, while *il15* showed no significant difference in expression between WT and *prkdc*^{-/-}/*il2rg*^{-/-} *Xenopus tropicalis* spleens. The expression levels of *il4*, *il9*, and *il21* were all relatively low in the spleens of both WT and *prkdc*^{-/-}/*il2rg*^{-/-} *Xenopus tropicalis*. Given that IL-2 is primarily produced by T cells, the significant reduction in *il2* expression in the spleen of *prkdc*^{-/-}/*il2rg*^{-/-} *Xenopus tropicalis* reflects the abnormal development of T cells in these frogs. (The response and modifications to this issue can be found in the manuscript on Line 356-358, Supplementary Fig. 25 for detailed information.)**

Responses Figure.8 The expression of five cytokines (*IL-2*, *IL-4*, *IL-9*, *IL-15*, and *IL-21*) in *prkdc*^{-/-}/*il2rg*^{-/-} (PI) *Xenopus tropicalis* is shown. *gapdh* was used as a RNA loading control.

4) Wild glasses with spectacular transparency have recently been reported, and the authors cited them in Reference 3 (Taboada, C. et al. Science. 2022;378(6626):1315-1320.) The frogs in Reference 3 are not *Xenopus tropicalis* and should be corrected accordingly. In the case of this glass frog, red blood cells in the blood circulation inhibit transparency. Therefore, in the case of *mitf*^{-/-} frogs, it would be interesting to know whether such inhibition of transparency by red blood cells was observed.

➤ We have corrected this error in the manuscript. Concerning the influence of red blood cells on the transparency of *mitf*^{-/-} *Xenopus tropicalis* skin, as indicated by the results in **Fig. 2B** of the manuscript, the skin of 1.5-month-old *mitf*^{-/-} *Xenopus tropicalis* remains transparent even when there is noticeable blood. This suggests minimal impact of red blood cells on the transparency of *mitf*^{-/-} *Xenopus tropicalis* skin. Additionally, we compared the skin of 8-month-old adult *mitf*^{-/-} *Xenopus tropicalis* with blood to those without blood, and the results indicate that blood has no impact on the transparency of adult *mitf*^{-/-} *Xenopus tropicalis* skin. (The response and modifications to this issue can be found in the manuscript on Line 204-207, Fig. 2C for detailed information.)

5) I may have missed it, but I couldn't find a description of the rearing of immunodeficient frogs. Quality control of rearing water is vital for immunocompromised aquatic animals. This information is essential for those who intend to experiment with these frogs.

➤ This issue has been addressed in the manuscript. (Refer to Line 529 – Line 532 for details.)

Minor comments.

1) The authors stated that the secretion of antimicrobial substances might be degraded due to failure of the formation of granular glands in the *mitf*^{-/-} frog. As the authors note, the *mitf* gene is involved in the formation of immunocompetent cells in humans (Ref. 55). Therefore, it is possible that *mitf*^{-/-} frogs functionally suppress the immune system and attenuate immunosuppressive effects. The authors need to discuss this possibility.

➤ *The mitf locus in Xenopus tropicalis is presumed to be similar to the Mitf locus in mammals, capable of transcribing multiple isoforms, each with distinct expression patterns and functions. In mammals, it is well-established that Mitf is highly expressed in mast cells, and various mast cell defects have been identified in several Mitf mutant mouse strains²¹. Mitf plays a crucial role in directing the differentiation of pre-BMPs (pre-basophil and mast cell progenitors) into mast cells while inhibiting basophil development through the repression of C/EBPα expression²¹. Mast cells are integral to the inflammatory process²¹. Consequently, the contribution of mitf knockout to immune rejection defects in Xenopus tropicalis is expected to be relatively minor. This viewpoint is supported by the significant infiltration of CD3⁺ cells in mitf^{-/-} Xenopus tropicalis after allogeneic skin transplantation (**Responses Figure.3**). However, the specific role of mitf in the immune system of Xenopus tropicalis remains a topic worthy of further exploration, and we plan to delve deeper into it when the opportunity arises. (This issue has been discussed in the manuscript, please refer to **Line 505 – Line 515** for details.)*

2) In Figure 3H, Mel is almost entirely missing from the field of view. It is better to expand the photographic area. Similarly, MG and S. bas in Fig. 3F and BC are shown in Fig. 3H, respectively.

➤ *This issue has been addressed in the manuscript. (Please refer to **Fig.3 and Supplementary Fig. 7** for details.)*

Reviewer #3

Ran et al., have developed a functionally nullifying biallelic *mitf* mutations that successfully created a colorless line of *Xenopus tropicalis*. This novel line displays transparent skin and visible internal organs within two months post-metamorphosis and lacks cutaneous melanophores throughout its lifespan. This newly developed amphibian model holds great potential for use in tumorous and developmental studies. Questions and suggestions concerning some details of the manuscript.

1. Include information from the literature about *Mitf* signaling pathways. For example, it is known that Wnt signaling (which must be included) which is very important in many biological processes plays a critical role in melanocyte regulation.

➤ *Indeed, as suggested by the reviewer, the manuscript requires supplementation with such information. The Mitf signaling pathways play a crucial role in melanocyte*

development, as discussed in the manuscript. β -catenin functions as a pivotal transcription factor in the WNT signaling pathway. WNT pathway activation in melanocyte development triggers β -catenin activation, subsequently activating MITF to govern neural crest cell development into melanoblast cells and establish the melanocyte lineage. The WNT/ β -catenin signaling pathway likely activates hair follicle melanocyte stem cells through MITF activation. Furthermore, during melanocyte development, post-translational modifications of MITF, such as phosphorylation at S73 and S409 by ERK and RSK via MAPK kinase pathway activation, play a crucial regulatory role. Phosphorylation at S73 is essential for ubiquitination and degradation of MITF. S73 phosphorylation can induce S69 phosphorylation by GSK3, and concurrent phosphorylation at both S73 and S69 activates a CRM1-dependent nuclear export signal, causing cytoplasmic translocation of MITF. In the absence of WNT pathway activation, GSK3 phosphorylates three C-terminal sites of MITF, promoting ubiquitination and degradation, thereby diminishing MITF's transcriptional activity. Upon WNT signaling pathway activation, the WNT machinery is sequestered in multivesicular bodies, resulting in MITF protein stabilization. The stabilized MITF subsequently induces the expression of late endosomal proteins, establishing a positive feedback loop involving MITF, multivesicular bodies, and WNT signaling during the proliferative stages of melanoma. GSK3 also enhances MITF stability by phosphorylating the S298 site. Additionally, PI3K, MAPK, and WNT signaling pathways not only collaboratively regulate MITF phosphorylation to control its function but also coordinate to regulate the expression of the transcription factor BRN2 (POU3F2), thus controlling MITF expression and function. In conclusion, the WNT signaling pathway primarily governs melanocyte development through MITF. Despite limited research on the interaction between Wnt and Mitf signaling pathways in *Xenopus tropicalis*, the conservation of melanocyte development in vertebrates suggests that, in *Xenopus tropicalis*, the Wnt signaling pathway predominantly regulates melanocyte development by controlling Mitf protein stability, nuclear localization, or protein expression levels (This issue has been discussed in the manuscript, please refer to Line 460 – Line 471 for details).

2. According to the literature GSK3, downstream from both the PI3K and Wnt pathways, and BRAF/MAPK signaling converges to control MITF. Did the authors test the interaction between Wnt and Wnt/GSK3? These possible interactions and regulation systems should be explain in the discussion.

➤ We appreciate the reviewer's suggestions, and we have incorporated this information into the discussion section of the manuscript. Unfortunately, our current research has not specifically investigated the impact of interactions between the PI3K, MAPK, WNT, and MITF signaling pathways on *Xenopus tropicalis* melanocyte development. As mentioned earlier, the PI3K, MAPK, and WNT signaling pathways primarily regulate melanocyte (or melanoma cell) development through the modulation of MITF phosphorylation and MITF protein expression. However, most experimental data supporting these pathways have been obtained from humans, mice, and zebrafish. It is noteworthy that some of these regulatory mechanisms have not been confirmed in

the normal development of melanocytes. Therefore, in future studies, elucidating the detailed molecular mechanisms through which the Pi3k, MAPK, Wnt, and Mitf signaling pathways regulate pigment cell development in *Xenopus tropicalis* would undoubtedly be an intriguing endeavor. (This issue has been discussed in the manuscript, please refer to Line 460 – Line 471 for details).

3. A culture cell experiment should be included showing a gain-of-function situation of the *mitf*^{-/-}/*prkdc*^{-/-}/*il2rg*^{-/-}?

➤ Thank you for your suggestion. In this manuscript, functional validation of the immunodeficiency in *mitf*^{-/-}/*prkdc*^{-/-}/*il2rg*^{-/-} *Xenopus tropicalis* for tumor transplantation has been achieved. Evidence of the impact of gene knockout on the immune system is provided through changes in thymus and spleen tissue structure. Additionally, to further illustrate the effects of *prkdc*^{-/-}/*il2rg*^{-/-} knockout on the immune system in *Xenopus tropicalis*, we performed qPCR to assess the expression of T cell, B cell, and NK cell marker genes, as shown in **Responses Figure.7**. The qPCR results indicate a significant reduction in the expression of T cell, B cell, and NK cell marker genes after *prkdc*^{-/-}/*il2rg*^{-/-} knockout, suggesting abnormal development of these immune cells. We also used CD3 antibodies to examine the infiltration of recipient *mitf*^{-/-} and *mitf*^{-/-}/*prkdc*^{-/-}/*il2rg*^{-/-} frogs T cells after allogeneic skin transplantation, as shown in **Responses-Figure.3**. These results collectively confirm that *prkdc*^{-/-}/*il2rg*^{-/-} knockout indeed results in immunodeficiency. Given the confirmation of our conclusions from the experimental results, further relevant cell experiments were not provided. (The response and modifications to this issue can be found in the manuscript on Line 353-356, Line 381, Fig.8E, Supplementary Fig.23, and Supplementary Fig.24 for detailed information).

4. Are β -catenin or Wnt target genes affected in the RT-PCR analysis?

➤ The question of whether knocking out *mitf* in *Xenopus tropicalis* affects the expression of β -catenin or Wnt target genes is indeed intriguing. In another research project, we conducted Bulk RNA sequencing on the dorsal skin of 1-year-old WT and *mitf*^{-/-} *Xenopus tropicalis* (grouped in sets of three frogs, with two replicates each). We examined the expression of β -catenin and some Wnt target genes in these data. As shown in **Responses Table.1**, there were no significant changes in the expression of β -catenin and these Wnt target genes in the skin of *mitf*^{-/-} frogs compared to WT frogs. This indicates that the mutual regulation between Mitf and the Wnt/ β -catenin signaling pathway in *Xenopus tropicalis* requires further investigation. The significant downregulation of Mitf target genes suggests the failure of transcriptional regulation dependent on Mitf in the skin of *mitf*^{-/-} frogs, explaining the loss of melanocytes in these frogs. (We have discussed this in the manuscript, please refer to Line 460 – Line 471 for details.)

Responses Table.1 Differential expression of *Mitf* and *Wnt/β-catenin* target genes

genes	baseMean	log2FoldChange	lfcSE	stat	pvalue	padj	significant	
axin2	234.1942	0.216180429	0.309262	0.699021	0.484539128	0.985529336	normal	beta-catenin target genes
mf43	196.24482	0.154014637	0.248643	0.619422	0.535638451	0.999811137	normal	
znrf3	37.266284	-0.126140964	0.610897	-0.20648	0.836412278	1	normal	
lgr5	196.05846	0.818207109	0.386685	2.115952	0.034348863	0.359952049	normal	
lef1	275.29059	0.295292872	0.305767	0.965745	0.334171673	0.905925084	normal	
tcf7	38.655514	0.492428854	0.782726	0.629121	0.529270033	0.999795467	normal	
tcf7l1	416.47357	0.464709863	0.262075	1.773195	0.076196358	0.543528182	normal	
tcf7l2	552.01008	0.092597987	0.205431	0.450751	0.652169345	1	normal	
ctnnb1	3739.115	0.271157893	0.139223	1.947654	0.051456376	0.454211179	normal	beta-catenin
tyr	197.02352	-9.913650913	1.532618	-6.46844	9.90195E-11	1.57485E-08	down	Mitf target genes
tyrp1	1032.2675	-10.13675003	1.061582	-9.54872	1.31312E-21	8.74053E-19	down	
dct	754.27837	-7.063203105	1.417689	-4.98219	6.28673E-07	5.53849E-05	down	
pmel	2036.0729	-12.9882957	1.491463	-8.70843	3.08118E-18	1.31845E-15	down	
slc45a2	120.71928	-5.850193645	0.931018	-6.28366	3.30703E-10	4.64328E-08	down	

5. Other signals control MITF activity through posttranslational modification. The authors should mention this in the discussion as well.

➤ *This issue has been discussed in the manuscript, please refer to Line 460 – Line 471 for details.*

6. I would recommend to include in the citations the following papers: It will be helpful to include a possible signaling pathway for MITF regulation.

<https://doi.org/10.1073/pnas.1810498115> K Takeda, et al., Ser298 of MITF, a mutation site in Waardenburg syndrome type 2, is a phosphorylation site with functional significance. *Hum Mol Genet* 9, 125–132 (2000).

<https://www.pnas.org/doi/10.1073/pnas.1810498115> D Ploper, et al., MITF drives endolysosomal biogenesis and potentiates Wnt signaling in melanoma cells. *Proc Natl Acad Sci USA* 112, E420–E429 (2015).

RI Dorsky, DW Raible, RT Moon, Direct regulation of nacre, a zebrafish MITF homolog required for pigment cell formation, by the Wnt pathway. *Genes Dev* 14, 158–162 (2000).

Albrecht, L. V., Tejada-Muñoz, N., & De Robertis, E. M. (2021). Cell Biology of Canonical Wnt Signaling. *Annual review of cell and developmental biology*, 37, 369–389. <https://doi.org/10.1146/annurev-cellbio-120319-023657>

➤ *Indeed, this is a valuable suggestion. These references play a crucial role in elucidating the function of MITF, and we have cited them in the manuscript (Please refer to Line 460 – Line 471 for details).*

References

- 1 Seki, T., Kikuyama, S. & Yanaihara, N. Development of *Xenopus laevis* skin glands producing 5-hydroxytryptamine and caerulein. *Cell and tissue research* **258**, 483-489, doi:10.1007/bf00218860 (1989).
- 2 Seki, T., Kikuyama, S. & Yanaihara, N. In vitro development of *Xenopus* skin glands producing 5-hydroxytryptamine and caerulein. *Experientia* **51**, 1040-1044, doi:10.1007/bf01946912 (1995).
- 3 VANABLE, J. GRANULAR GLAND DEVELOPMENT DURING *XENOPUS LAEVIS* METAMORPHOSIS. *Developmental biology* **10**, 331-357, doi:10.1016/0012-1606(64)90049-1 (1964).

- 4 Reilly, D., Tomassini, N. & Zasloff, M. Expression of magainin antimicrobial peptide genes in the developing granular glands of *Xenopus* skin and induction by thyroid hormone. *Developmental biology* **162**, 123-133, doi:10.1006/dbio.1994.1072 (1994).
- 5 Fukuzawa, T. Unusual leucophore-like cells specifically appear in the lineage of melanophores in the periodic albino mutant of *Xenopus laevis*. *Pigment cell research* **17**, 252-261, doi:10.1111/j.1600-0749.2004.00135.x (2004).
- 6 Yasutomi, M. & Hama, T. Electron microscopic study on the xanthophore differentiation in *Xenopus laevis*, with special reference to their pterinosomes. *Journal of ultrastructure research* **38**, 421-432, doi:10.1016/0022-5320(72)90080-9 (1972).
- 7 Weiher, H., Noda, T., Gray, D. A., Sharpe, A. H. & Jaenisch, R. Transgenic mouse model of kidney disease: Insertional inactivation of ubiquitously expressed gene leads to nephrotic syndrome. *Cell* **62**, 425-434, doi:[https://doi.org/10.1016/0092-8674\(90\)90008-3](https://doi.org/10.1016/0092-8674(90)90008-3) (1990).
- 8 Spinazzola, A. *et al.* MPV17 encodes an inner mitochondrial membrane protein and is mutated in infantile hepatic mitochondrial DNA depletion. *Nature Genetics* **38**, 570-575, doi:10.1038/ng1765 (2006).
- 9 Parini, R. *et al.* Glucose metabolism and diet-based prevention of liver dysfunction in MPV17 mutant patients. *Journal of Hepatology* **50**, 215-221, doi:10.1016/j.jhep.2008.08.019 (2009).
- 10 Martorano, L. *et al.* The zebrafish orthologue of the human hepatocerebral disease gene MPV17 plays pleiotropic roles in mitochondria. *Dis Model Mech* **12**, doi:10.1242/dmm.037226 (2019).
- 11 D'Agati, G. *et al.* A defect in the mitochondrial protein Mpv17 underlies the transparent casper zebrafish. *Dev Biol* **430**, 11-17, doi:10.1016/j.ydbio.2017.07.017 (2017).
- 12 Bian, C. *et al.* Genome and Transcriptome Sequencing of casper and roy Zebrafish Mutants Provides Novel Genetic Clues for Iridophore Loss. *Int J Mol Sci* **21**, doi:10.3390/ijms21072385 (2020).
- 13 Subkhankulova, T. *et al.* Zebrafish pigment cells develop directly from persistent highly multipotent progenitors. *Nature Communications* **14**, doi:10.1038/s41467-023-36876-4 (2023).
- 14 Goding, C. & Arnheiter, H. MITF-the first 25 years. *Genes & development* **33**, 983-1007, doi:10.1101/gad.324657.119 (2019).
- 15 Ran, R. *et al.* Disruption of tp53 leads to cutaneous nevus and melanoma formation in *Xenopus tropicalis*. *Molecular oncology* **16**, 3554-3567, doi:10.1002/1878-0261.13301 (2022).
- 16 Zhao, H., Han, D., Dawid, I., Pieler, T. & Chen, Y. Homeoprotein hhex-induced conversion of intestinal to ventral pancreatic precursors results in the formation of giant pancreata in *Xenopus* embryos. *Proceedings of the National Academy of Sciences of the United States of America* **109**, 8594-8599, doi:10.1073/pnas.1206547109 (2012).
- 17 Zhang, T., Guo, X. & Chen, Y. Retinoic acid-activated Ndr1a represses Wnt/ β -catenin signaling to allow *Xenopus* pancreas, oesophagus, stomach, and duodenum specification. *PloS one* **8**, e65058, doi:10.1371/journal.pone.0065058 (2013).
- 18 Osusky, M. *et al.* Transgenic plants expressing cationic peptide chimeras exhibit broad-spectrum resistance to phytopathogens. *Nature biotechnology* **18**, 1162-1166, doi:10.1038/81145 (2000).
- 19 Yeo, G. *et al.* Phylogenetic and evolutionary relationships and developmental expression patterns of the zebrafish twist gene family. *Development genes and evolution* **219**, 289-300, doi:10.1007/s00427-009-0290-z (2009).

- 20 Enyedi, B., Kala, S., Nikolich-Zugich, T. & Niethammer, P. Tissue damage detection by osmotic surveillance. *Nature cell biology* **15**, 1123-1130, doi:10.1038/ncb2818 (2013).
- 21 Kim, S., Song, H. S., Yu, J. & Kim, Y. M. MiT Family Transcriptional Factors in Immune Cell Functions. *Mol Cells* **44**, 342-355, doi:10.14348/molcells.2021.0067 (2021).

REVIEWERS' COMMENTS:

Reviewer #1 (Remarks to the Author):

In general, the authors have addressed my concerns and either added the data/comment to the revision or shown that the experiment was not producible. I do have one minor concern that requires edits to the text only.

Minor Comments:

1. Line 269-270. Given the current RT-PCR analysis, it is unclear if the absence of mature melanocytes is due to a failure to maintain and survive melanoblasts or due to another defect such as a failure to specify or differentiate. I recommend changing the phrasing to avoid any unresolved conclusions.

Reviewer #2 (Remarks to the Author):

I am very much looking forward to the future development of this research, which challenged the creation of a new animal model through transparency and modification of the immune function of *Xenopus tropicalis*. The authors have substantially revised the manuscript in response to the comments of the reviewers, including myself, and I have carefully reviewed the manuscript. The authors have been thorough in responding to all comments sincerely addressed. We believe that this content will attract the attention of a wide range of researchers. The visual abstracts added to the revised manuscript make the overall experimental scheme more accessible to those who do not specialize in amphibians. During the review of this paper, I found that the following related new papers have been published. (Generation of translucent *Xenopus tropicalis* through triple knockout of pigmentation genes. *Development, Growth & Differentiation*, 65(9), 591 -598. <https://doi.org/10.1111/dgd.12891>) There is no risk that the authors' work will be neglected by this report. The reader will appreciate it if this related paper is properly cited. Therefore, I recommend that the authors cite this appropriately in the final manuscript.

Reviewer #3 (Remarks to the Author):

Ran et al., have developed a transgenic line by triple knockout of *mitf*^{-/-}/*prkdc*^{-/-}/*il2rg*^{-/-} that successfully created a transparent and immunodeficient *Xenopus tropicalis*. The authors addressed each of my questions very well. Therefore, I consider the manuscript appropriate for publication for its implications in cancer research.

Response to Referees

We thank the referees for their comments and suggestions. We have taken these comments into account in the revision of the manuscript.

Reviewer #1 (Remarks to the Author):

In general, the authors have addressed my concerns and either added the data/comment to the revision or shown that the experiment was not producible. I do have one minor concern that requires edits to the text only.

Minor Comments:

1. Line 269-270. Given the current RT-PCR analysis, it is unclear if the absence of mature melanocytes is due to a failure to maintain and survive melanoblasts or due to another defect such as a failure to specify or differentiate. I recommend changing the phrasing to avoid any unresolved conclusions.

➤ *We Thank you for pointing out these issues. They have been addressed and corrected in the manuscript. (Please refer to Line 271-273 for details.)*

Reviewer #2 (Remarks to the Author):

I am very much looking forward to the future development of this research, which challenged the creation of a new animal model through transparency and modification of the immune function of *Xenopus tropicalis*. The authors have substantially revised the manuscript in response to the comments of the reviewers, including myself, and I have carefully reviewed the manuscript. The authors have been thorough in responding to all comments sincerely addressed. We believe that this content will attract the attention of a wide range of researchers. The visual abstracts added to the revised manuscript make the overall experimental scheme more accessible to those who do not specialize in amphibians. During the review of this paper, I found that the following related new papers have been published. (Generation of translucent *Xenopus tropicalis*

through triple knockout of pigmentation genes. *Development, Growth & Differentiation*, 65(9), 591 -598. <https://doi.org/10.1111/dgd.12891>) There is no risk that the authors' work will be neglected by this report. The reader will appreciate it if this related paper is properly cited. Therefore, I recommend that the authors cite this appropriately in the final manuscript.

➤ *We Thank you for pointing out this issue. It has been addressed and corrected in the manuscript. (Please refer to Line 113-117 for details.)*

Reviewer #3 (Remarks to the Author):

Ran et al., have developed a transgenic line by triple knockout of *mitf*^{-/-}/*prkdc*^{-/-}/*il2rg*^{-/-} that successfully created a transparent and immunodeficient *Xenopus tropicalis*. The authors addressed each of my questions very well. Therefore, I consider the manuscript appropriate for publication for its implications in cancer research.

➤ *We Thank you for your comments.*